


# The benefits of pre- and postprocessing streamflow forecasts for an operational flood-forecasting system of 119 Norwegian catchments

Trine J. Hegdahl[1], Kolbjørn Engeland[1,2], Ingelin Steinsland[3], Andrew Singleton[4]

[1]Norwegian Water Resources and Energy Directorate, Hydrological Modelling, 0301 Oslo, Norway
[2]University of Oslo, Department of Geosciences, 0316 Oslo, Norway
[3]Norwegian University of Science and Technology, Department of Mathematical Sciences, 7034 Trondheim, Norway
[4]Norwegian Meteorological Institute, 0313 Oslo, Norway

*Correspondence to*: Trine J. Hegdahl (tjh@nve.no)

**Abstract**. The novelty of this study is to evaluate the univariate and the combined effects of including both precipitation and temperature forecasts in the preprocessing together with the postprocessing of streamflow for forecasting of floods as well as all streamflow values for a large sample of catchments. A hydrometeorological forecasting chain in an operational flood forecasting setting with 119 Norwegian catchments was used. This study evaluates the added value of pre- and postprocessing methods for ensemble forecasts in a hydrometeorological forecasting chain in an operational flood forecasting setting with 119 Norwegian catchments. Two years of ECMWF ensemble forecasts of temperature (T) and precipitation (P) with a lead-time up to 9 days were used to force the operational hydrological HBV model to establish streamflow forecasts. Two approaches to preprocess the temperature and precipitation forecasts were tested. 1) An existing approach applied to the gridded forecasts using quantile mapping for temperature and a Bernoulli-gamma distribution for precipitation. 2) Bayesian model averaging (BMA) applied to catchment average values of temperature and precipitation. BMA was also used for postprocessing catchment streamflow forecasts. Ensemble forecasts of streamflow were generated for a total of fourteen schemes based on combinations of raw, preprocessed, and postprocessed forecasts in the hydrometeorological forecasting chain. The aim of this study is to assess which pre- and postprocessing approaches should be used to improve streamflow and flood forecasts and look for regional or seasonal patterns in preferred approaches.

The forecasts were evaluated for two datasets: i) all streamflows and ii) flood events with streamflow above mean annual flood. Evaluations were based on reliability, continuous ranked probability score (CRPS) and -skill score (CRPSS). For the flood dataset, the critical success index (CSI) was used. Evaluations based on all streamflow data showed that postprocessing improved the forecasts only up to a lead-time of two to three days, whereas preprocessing T and P using BMA improved the forecasts for 50% - 90% of the catchments beyond three days lead-time. However, for flood events, the added value of pre- and postprocessing is smaller. Preprocessing of P and T gave better CRPS for marginally more catchments compared to the other schemes.





Based on CSI, we found that many of the forecast schemes perform equally well. Further, we found large differences
in the ability to issue warnings between spring and autumn floods. There was almost no ability to predict autumn floods
beyond 3 days, whereas the spring floods had predictability up to 9 days for many events and catchments. The results
indicate that the ensemble forecasts have problems in predicting correct autumn precipitation, and the uncertainty is
larger for heavy autumn precipitation compared to spring events when temperature driven snow melt is important. To
summarize we find that the flood forecasts benefit from most pre- and postprocessing schemes, although the best
processing approaches depend on region, catchment, and season, and that the processing scheme should be tailored to
each catchment, lead time, season, and the purpose of the forecasting.

## 1    Introduction

Floods can have severe economic, personal, and social costs. Early warnings based on flood forecasts enable both the
management authorities and the public to take necessary measures to reduce the impact of floods (e.g., UNISDRI,
2004, Pappenberger et al., 2015). However, predicting the future is adhered with uncertainty. Attaching the forecast
uncertainty to a predicted flood level adds value for many end users allowing them to do risk evaluation in light of
their often-unique circumstances, and thus take measures that are most appropriate and cost effective for them.
In the hydro-meteorological forecasting chain there are multiple sources to uncertainty. There is uncertainty in
observations, initial conditions, forcing data, model description, and model parameters (e.g., Buizza et al., 1999; Zappa
et al., 2011). For flood forecasting an important source of uncertainty and errors are the forcing in the forecasting
period, i.e. precipitation and temperature weather forecasts (e.g. Zappa et al., 2011), and this is the focus of this paper.
From weather prediction systems it is known that small changes in the initial conditions will affect atmospheric
trajectories and future weather predictions (e.g., Lorenz, 1969; Buizza, 2008). To capture the uncertainty in weather
prediction caused by initial conditions and model parametrization, ensemble prediction systems (EPS) were developed
as early as the 70s (Leith, 1974). The use of meteorological ensembles as input to hydrological models is one approach
to achieve probabilistic streamflow forecasts, and thereby provide a probability of the forecasted flood to exceed a
given level (Buizza, 2008).
Today, ensemble weather forecasts are available as operational services, and using these for hydrological forecasts
have been studied in the literature, see e.g., Cloke and Pappenberger (2009) and Wetterhall et al. (2013). To get
unbiased and reliable hydrological forecasts, preprocessing (applied to the meteorological forcing) and/or
postprocessing (applied to the hydrological output) techniques are needed. Several processing methods are proposed
in literature, see e.g., Vannitsem et al. (2018) for an overview. For a national or regional flood forecasting service, a
large number of catchments with different hydrological processes and regimes are considered. Therefore, to assess the
added value of pre- and postprocessing, a dataset from a large number of catchments that well represent the variability
on hydrological processes is needed to provide robust conclusions. In addition, it is important to assess (and compare)
the performance of flood forecast, not all streamflow values, for different pre- and postprocessing schemes. In most
papers, ensemble forecasts of all streamflow values for one or a small number of catchments are evaluated. This paper
aims to fill two knowledge gaps: 1) To gain understanding of the differences in quality for pre and/or post processing
method for a range of catchments, and 2) The assess the quality of pre- and postprocessing for flood forecasts.





Reliability and accuracy are key characteristics used to measure the quality of ensemble forecasts. A reliable forecast
is statistically calibrated (e.g., for 90% of the forecasts, the observations are within the 90% prediction interval). Raw
ensemble forecasts are rarely reliable in this sense. The discrepancy between the weather predictions and point
measurements shows that forecast ensembles are often biased and underdispersive (Gneiting et al., 2005). A lack of
dispersion in global meteorological ensembles is most evident for the shortest lead times and can be explained by
slower growth rates of the perturbations in the ensemble prediction system compared to those of an instable "true"
atmosphere (Hamill and Colucci, 1997). To correct for bias and underdispersion in the ensemble system, different
statistical postprocessing approaches are applied to achieve calibrated ensembles. Li et al. (2017) and Vannitsem et al.
(2018) provide a comprehensive review of processing techniques, both parametric approaches relying on parametric
probability distributions, for example Bayesian model averaging (BMA) and non-homogeneous Gaussian regression
(NGR), and nonparametric approaches like quantile regression and ensemble error dressing methods. Raferty et al.
(2005) introduced BMA to the atmospheric community as a statistical method to achieve calibrated and sharp forecasts,
and the method has since been widely used within the community (Fraley et al., 2010). More recently studies that use
BMA for postprocessing to improve streamflow forecasts have been carried out. For example, Madadgar et al. (2014)
used copula embedded BMA for postprocessing streamflow forecasts and improved the forecasts compared to quantile
mapping techniques. Jha et al. (2018) demonstrated the use of BMA to remove bias and reduce errors in the
precipitation forecasts responsible for a flood event. NGR accounts for the errors in the mean, but unlike an ordinary
regression, the error variance is not assumed to be constant, but rather to vary linearly with respect to the ensemble
variance (Wilks and Hamill, 2007; Gneiting et al., 2005). Quantile regression applied to ensemble forecasts was
introduced by Bremnes (2004) and was first used to correct precipitation forecasts. The method can be viewed as a
non-parametric counterpart to NGR, where the predictive probability distribution is described by a set of quantiles.
Linear regression is used to describe the relationship between the observations and the forecasts, and the regression
parameters are specific for each quantile. There are variations of most methods, and ensemble dressing is one that has
both parametric and non-parametric approaches. Roulston and Smith (2003) suggested a non-parametric kernel
dressing method, where the kernel represents a distribution of errors from previous forecasts, which is applied to each
member of the ensemble. Wang and Bishop (2005) extended this idea and suggested the use of a parametric dressing
method of Gaussian kernels where the parameters were estimated by the training data.
Previous studies have analyzed the effects of both pre- and postprocessing on short- to medium-range ensemble
streamflow forecasts (e.g., Zalachori et al., 2012; Roulin and Vannitsem, 2015; Benninga et al., 2017, Sharma et al.,
2018). Few studies include preprocessing of temperature. Verkade et al. (2013), Benninga et al. (2017), and Hegdahl
et al. (2019) all applied variations of quantile mapping techniques to calibrate the temperature forecasts, whereas
Zalachori et al. (2012) applied an analog approach. Hegdahl et al. (2019) showed that in catchments with seasonal
snow cover, temperature calibration is important for improved streamflow forecasts Variations of logistic regression
approaches are most common in the studies that preprocessed precipitation (Verkad et al., 2013; Roulin and Vannitsem
et al., 2015; Benninga et al., 2017; Sharma et al., 2018). One exception is the analog approach applied by Zalachori et
al. (2012). A larger variety of approaches are used to postprocess streamflow; Bayesian processing (Reggiani et al.,
2009), Bayesian model averaging including multi-model approaches (Rings et al. 2012; Parish et al. 2012; Xu et al.,
2019), variations of quantile regression (Bogner et al., 2016; Benninga et al., 2017; Sharma et al., 2018), extended
logistic regression (Fundel and Zappa 2011), and ensemble model output statistics (Roulin and Vannitsem 2015). Some


key findings are that calibrated precipitation forecasts do not necessarily lead to calibrated streamflow forecasts
(Zalachori et al., 2012; Verkade et al., 2013; Benninga et al., 2017). Postprocessing alone is the simplest way to
improve forecasting performance (Zalachori et al., 2012; Sharma, 2018), but not always with a significant improvement
(Benninga et al., 2017). Preprocessing the meteorological forcing is important for forecasting high streamflow since
errors from the meteorological model are dominant in this case (Benninga et al., 2017). Preprocessing has the highest
skill improvement in the warm season, whereas postprocessing is the most effective in the cold season with snow cover
(Sharma et al., 2018). This summary indicates that the relative importance of pre- and postprocessing depends on
factors including lead time, streamflow magnitude and season.
From the literature on short- to medium range streamflow forecasts we have identified two studies investigating the
combined effect of preprocessing temperature and precipitation as well as postprocessing the streamflow (Benninga et
al., 2017; Zalachori et al., 2012). However, neither of these two studies consider the impacts of such pre- and
postprocessing strategies on the forecasting of flood events directly. Zalachori et al. (2012) assess the performance for
all streamflows and Benninga et al. (2017) assess the performance for, not necessarily flood inducing, high flows. In
Benninga et al. (2017) the forecasts are evaluated for only one catchment, and the author acknowledge that more
catchments are needed to verify the generality of their results. A 'large catchment sample' is needed to draw robust
conclusions in such studies (Gupta et al., 2014).
The two unique contributions of our study are to (i) evaluate the univariate and the combined effects of including both
precipitation and temperature forecasts in the preprocessing together with the postprocessing of streamflow for
forecasting of both floods as well as all streamflow values, and to (ii) perform the evaluation for a large catchment
sample. Evaluating the performance of processing approaches on flood forecasts is critical, since we can expect the
processing approaches to be less efficient for extreme and often unique flood events. Using a large catchment sample
allows us to investigate how the performance depends on both climatological and physiographic catchment
characteristic and to draw more robust conclusions. Furthermore, this comprehensive evaluation is performed for lead
times ranging from 1 to 9 days and the performance is assessed for different seasons.
Following the works cited above, the working hypothesis of this paper is that pre- and/or postprocessing improves
streamflow forecasts, but that the improvement might differ between catchments and between events. The main
objective of this study is to assess the potential improvements in flood forecasts by combining pre- and postprocessing
for a variety of catchments. We addressed the following questions:
1.  Which pre- and postprocessing approaches should be used in the hydrometeorological forecasting chain to
30       improve streamflow forecasts with an emphasis on flood forecasting?

2.  Are there regional or seasonal patterns in preferred pre- and postprocessing approaches?
In this study, we applied and evaluated the different processing schemes within the operational flood forecasting setup
used by the Norwegian flood forecasting service. The different schemes were tested for 119 catchments that vary in
climatology, catchment characteristics, and hydrological regimes. The large number of flood events and catchments
allowed us to provide robust assessments of the performance of the different schemes under different flood conditions.


**2    Study Area, Hydrological Model and Data**
**2.1    Area**
The west coast of Norway forms a topographical barrier for the westerlies. The resulting orographic enhancement of
precipitation makes this area one of the wettest parts of Europe, with an annual precipitation of around 4000mm,
whereas the driest regions in the rain shadow of the mountains have annual precipitation of around 400mm (Hanssen-
Bauer, 2017). The temperature depends both on latitude, altitude, and distance from the coast. The catchments belong
to Köppen-Geiger climate classes ranging from subarctic in the north and at high elevations, to temperate in the coastal
areas (according to the Köppen-Geiger climate classes as defined in Peel et al., 2007).
The spatial patterns of mean precipitation explain most of the spatial patterns in mean runoff. The seasonal variation
in runoff depends on seasonal variations in both temperature and precipitation. There are two basic runoff regimes in
Norway. For coastal regions with a temperate climate, the highest flows occur during autumn and winter due to heavy
rainfall.  For inland regions with a sub-arctic or arctic climate, prolonged periods of winter temperatures below zero
ᵒC result in a seasonal snow storage, winter low flow, and high streamflow during spring due to snowmelt. There are,
however, many possible transitions between these two basic patterns (e.g., Gottschalk et al., 1979).
The study area consists of 119 catchments distributed all over Norway (Fig 1). All selected catchments are part of the
operational flood forecasting system and are mostly unregulated, with a large variation in size (3 to 15447 $km^2$) and
elevation (103 to 2284 meter above sea level [m.a.s.l.]). Six catchments are presented in more detail, the location of
these are indicated in Fig 1 and some key characteristics in table 1. The three first catchments are used as examples of
changes in reliability, depending on processing methods, datasets, and lead time.  The three last catchments are used
to illustrate streamflow forecasts estimated by different processing approaches for three different flood events.
**2.2    Hydrological Model**
We used the Hydrologiska Byråens Vattenbalance (HBV) model (Bergstrøm, 1974; Beldring, 2006; Sælthun, 1996)
that is used in the operational flood forecasting service at the Norwegian Water resources and Energy Directorate
(NVE). The HBV model is a conceptual model where the vertical structure of the model includes a snow routine, a
soil moisture routine, and a response function that consists of two tanks. Quick runoff is represented by a non-linear
tank, whereas slow runoff is represented by a linear tank. The model divides each catchment into 10 elevation zones
where each represents 10% of the catchment area. Catchment average temperature and precipitation are elevation
adjusted using a catchment specific lapse-rate to attain one representative precipitation and temperature value for each
elevation zone. The Nash-Sutcliff efficiency (Nash and Sutcliffe, 1970) and volume bias are used as calibration metrics.
The calibration period, 1996-2012, gives a mean Nash-Sutcliffe 0.77 for all 119 catchments, with zero volume bias.
The validation period, 1980-1995, shows mean Nash-Sutcliffe 0.73, with a mean volume bias of 5% (Ruan, 2016).
**2.3    Data**
**2.3.1    Meteorological observation SeNorge v1.1**
We used the gridded daily temperature and precipitation data from SeNorge v 1.1 that covers all of Norway with a 1x1
km grid size. The interpolation of observations to the grid is based on measured values at approximately 400



meteorological stations for precipitation, and 240 stations for temperature. Residual kriging is used for spatial
interpolation of de-trended temperature values (Tveito, 2007; Mohr, 2008). Temperature is detrended by adjusting
station data to sea level using a standard temperature lapse rate of 0.65 °C/100m. Triangulation is used for the spatial
interpolation of precipitation (Tveito, 2007; Mohr, 2008). The precipitation is further elevation corrected, using a
constant increase of 10% per 100 m beneath 1000 m.a.s.l, and 5% per 100 m above 1000 m.a.s.l. (Tveito et al., 2005).
**2.3.2    Meteorological forecasts ECMWF ENS.**
The temperature and precipitation forecasts used in the hydrological simulations of this study were taken from the
European Center of Medium-Range Weather Forecast (ECMWF) forecast ensembles (ENS). ENS provides an
ensemble of 51 members, with a forecasting period of 246 hours. The generation of the members of the ensemble is
done by adding small perturbations, which represent the uncertainty in the observations, to the forecast initial
conditions. Further, the uncertainty associated with the model physics is represented by perturbing the physics
tendencies that come from the parametrizations and each member is perturbed individually. This method is known as
the Stochastically Perturbed Parametrization Tendencies (SPPT) scheme and improves the forecasts giving a much
better spread-error relationship compared to initial condition perturbations alone. A detailed description of the
ECMWF ENS system is provided in e.g. Buizza et al. (1999) and Persson (2015). The grid resolution of the model
forecasts used implemented in this study is 0.25° (i.e. model cycles/versions 40r1, and 41r1 (ECMWF, 2018b)). The
variables used for the hydrological modelling are the 2-meter temperature and the accumulated precipitation
aggregated to catchment daily (06:00-06:00) mean values.
**2.3.3    Streamflow reference simulations**
The streamflow measurements from the NVE database (https://www.nve.no/hydrology/ ) were used as a reference for
the hydrological model calibration. To evaluate the streamflow forecasts, we used simulated streamflow created by
running the hydrological model with SeNorge temperature and precipitation as forcing. Using this approach, we
isolated the effect of the uncertainty in the weather forecasts, and we could ignore uncertainty in hydrological model
parameters, parametrizations, and calibration.
**2.4    Study period**
The years 2014 and 2015 were chosen as the study period since several large floods affected rivers in most parts of the
country during this two-year period (Figure 1). In May 2014 there were large snowmelt floods in central and eastern
parts of Norway (affecting the Lågen, Glomma, and especially the unregulated Trysilelva catchments). In October
2014 western Norway was hit by an atmospheric river (a narrow plume of high moisture content transported from the
tropical and extratropical latitude towards the poles, see e.g., Zhu and Newell 1998), which led to flooding of multiple
rivers. Atmospheric rivers are responsible for extreme precipitation events when the moist air masses are
orographically lifted at topographical barriers like the west coast of Norway (e.g., Stohl et al., 2008). In July 2015 there
were snowmelt floods in Oppland (central eastern Norway), and in September 2015 an extratropical cyclone, *Petra*,
caused floods in Southern Norway. In early October 2015, a cyclone, *Roar*, that caused floods in Trøndelag and
Nordland and in early December a cyclone, *Synne,* caused floods in several catchments in south-west Norway, some
exceeding the 200-year return level.





During the study period 2014 and 2015, floods did not occur in all catchments; hence, the number of catchments used
in the flood evaluation analysis was reduced to 80. We still used all 119 catchments when evaluating the performance
for all streamflow values.
**3    Pre- and postprocessing**
We applied processing steps to both the weather input to the hydrological model and its streamflow output. To
distinguish the different processing steps, we refer to preprocessing as corrections schemes applied to temperature and
precipitation ensembles, and postprocessing as corrections applied to the hydrological ensembles.
**3.1    Processing chain**
The temperature and precipitation forecast data from ECMWF were prepared by aggregating the variables from hourly
to a daily time step. Thereafter the horizontal resolution was changed using nearest neighbor interpolation to a 1×1 km
grid, equal to the SeNorge grid. For the temperature forecasts, a standard elevation adjustment of 0.65℃/100m was
applied to account for the elevation differences between the original and the seNorge grid. Finally, the temperature
and precipitation forecasts were aggregated to average values for each catchment. We used the ECMWF forecasts from
2014 and 2015 to force the hydrological model, which enabled a retrospective evaluation of the daily streamflow
forecasts for almost two years. The unprocessed daily forecasts for each catchment are referred to as $Traw_{t,l,s,m}$ and
$Praw_{t,l,s,m}$ where $t$ is issue time, $l$ is lead time, $s$ is catchment and $m$ is ensemble member. For temperature and
precipitation forecasts, two different preprocessing approaches were chosen, a grid calibration (CAL) producing the
ensembles $Tcal_{t,l,s,m}$ and $Pcal_{t,l,s,m}$, and Bayesian model averaging (BMA) producing the ensembles $Tbma_{t,l,s,m}$ and
$Pbma_{t,l,s,m}$. For postprocessing of streamflow, we used BMA to create $Qbma_{t,l,s,m}$. For all approaches, the processing
was applied to each issue date, $t$, lead time $l$ and catchment, $s$, independently. To improve readability, $t,l,s,m$ is
suppressed in the remainder of this paper. We evaluated all combinations of $Tcal$ and $Pcal$ together with $Traw$ and
$Praw$, as well as all combinations of $Tbma$ and $Pbma$ together with $Traw$ and $Praw$. $Tcal$ and $Pcal$ was not combined
with $Tbma$ and $Pbma$. The seven combinations of temperature and precipitation were run through the hydrological
model resulting in seven unprocessed streamflow forecasts ($Qraw$). Thereafter, postprocessing the raw forecasts
resulted in seven streamflow forecasts ($Qbma$), which could be compared to $Qraw$ to establish the effect of
postprocessing. Figure 2 provides an overview of the complete processing chain. More detailed presentation of each
step in the processing chain follows.
Different observational reference data and periods were the basis for the different processing techniques. An overview
of the variables, resolution, and data used for training are presented in Table 2 and details are provided in the following
subsections.
**3.2    Grid calibration**
The Norwegian Meteorological Institute (MET Norway) uses grid calibration approaches to improve ensemble
forecasts that are used for the operational national weather forecasts published at yr.no (methods available at
https://github.com/metno/gridpp/). We have rerun the preprocessing of the daily ensemble forecasts of temperature



and precipitation between 2014 and 2015, using the operational processing methods at that time. In the following text
these are referred to with the subscript *cal*. All calibration parameters were provided by MET Norway.
For the grid calibration methods, we applied the same corrections to each ensemble member. The ordering of members
was therefore kept. Thereby consistency between the calibrated temperature and precipitation members was ensured
and the temporal profile was preserved, which is important for the hydrological modelling.
### 3.2.1  Temperature calibration (Tcal)
Quantile mapping (Seierstad, 2016; Bremnes, 2007) was used to remove biases in the temperature forecasts by moving
the ensemble (ENS) forecast climatology closer to the observed climatology. MET Norway used Hirlam (Bengtsson
et al., 2017) temperature forecast at a $4\times4$ km$^2$ grid, as a reference for parameter estimation used to calibrate the
ECMWF ENS. Hirlam was the operational regional model at the time and is suitable as a reference since it provides a
continuous field covering all of Norway at a sub daily time step. Hirlam gives a higher skill and is less biased than
ENS when they are compared to point observations (Engdahl et al., 2015). To establish the calibration parameters,
MET Norway used both ENS reforecasts (Owens, 2018) and Hirlam data from July 2006 to December 2011
interpolated to a $5\times5$ km$^2$ grid. The ENS reforecast is a 5-member ensemble generated from the same model cycle
(40r1 and 41r1) as the operational ENS forecasts. For each grid cell, quantile transformation coefficients unique for
each month of the year, were determined by using data from a three-month window centered on the target month, e.g.
the May analysis consists of April, May, and June (Seierstad, 2017). The coefficients were estimated by mapping the
first 24 hours of the forecasts. A 1:1 extrapolation was used for forecasts outside the range of observation. In this study
we used the quantile transformation coefficients estimated by MET Norway. This enabled us to establish a
retrospective calibration of the temperature ensemble forecasts.
### 3.2.2  Precipitation calibration (Pcal)
To account for the intermittent nature of daily precipitation, a Bernoulli-Gamma distribution was used to calibrate the
precipitation forecasts. Precipitation observations from around 200 WMO stations in Norway are used to establish the
parameters of the Bernoulli-Gamma model. All parameters in the Bernoulli-Gamma model depend on lead-time, but
independent of location and issue date.
The probability mass for zero precipitation was specified by logistic regression. Both the cube transformed, and the
untransformed ensemble means and the fraction of ensemble members with precipitation higher than 0.5mm were used
as predictors. A total of four parameters were estimated in the logistic regression model. The precipitation amounts
were modelled by a gamma distribution The cube root of the forecast ensemble mean is used as a predictor in a model
with two parameters to fit the mean, whereas the untransformed forecast ensemble mean is used as a predictor in a
model with two parameters are used to fit the standard deviation. MET Norway provided the parameters that were used
at the issue time of the precipitation forecasts, and we applied them for a retrospective calibration of the precipitation
ensemble forecasts.
### 3.3  Bayesian Model averaging


1 Bayesian model averaging (BMA) aims to correct dispersion errors in a bias corrected ensemble (Raferty et al 2005).

2 For each lead time, BMA uses a mixture distribution, where for an ensemble with $M$ members, the density function

3 conditioned on all ensemble members is the weighted average of kernels for each member $m$. The preprocessed

4 meteorological ensembles were established by randomly drawing $M$ realizations from the mixture distribution

5 estimated by BMA. The kernel, for the quantity one wishes to forecast, $y$, is denoted by $f_\theta(y|x_m)$ where $x_m$ is the raw

6 forecast's ensemble member $m$ and $\theta$ are parameters of the kernel pdf $f$. The probability density function conditioned

7 on all $M$ ensemble members is the weighted average of the pdf for each member:

$$f(y|x_1,\dots,x_M) \sim \sum_{m=1}^{M} w_m f_\theta(y|x_m), \tag{1}$$

8 where $\sum_{m=1}^{M} w_m = 1$ and the weights are interpreted as the posterior probabilities of each ensemble member. The

9 ensembles in this paper are based on ECMWF ENS which are considered exchangeable, and weights and parameters

10 can be constrained to be equal for all members (Fraley et al 2010). For each issue date we used the previous $n$ days of

11 ensemble forecasts and reference observations to estimate the parameters in the kernel. To account for the specific

12 properties of temperature, precipitation and streamflow, different kernel distributions were used, the details are

13 provided below.

### 3.3.1 BMA for temperature (Tbma)

15 We followed Raferty et al (2005) and used a Normal distribution as the kernel for the temperature BMA models. Since

16 the temperature ensemble forecasts were not already bias corrected, the mean is specified as $a_0 + a_1 T_{raw,m}$ , where

17 $T_{raw,m}$ is the temperature forecast for ensemble member $m$ and $a_0$ and $a_1$ are regression parameters that account for any

18 bias. The parameters are specific for each catchment, issue date and lead time and are the same for all ensemble

19 members.

$$f(T_{bma}|T_{raw,m}) \sim N(a_0 + a_1 T_{raw,m}, \sigma^2), \tag{2}$$

20 To estimate the parameters, the catchment average temperatures from SeNorge were used as a reference.

### 3.3.2 BMA for precipitation (Pbma)

22 We followed Sloughter et al (2007) who proposed a Bernoulli-gamma distribution as kernel in the BMA precipitation

23 models to establish $P_{bma}$.

$$
\begin{aligned}
f(P_{bma}|P_{raw,m}) = \ &f(P_{bma}=0|P_{raw,m})I_{\{P_{bma}=0\}} \\
&+ f(P_{bma}>0|P_{raw,m})h(P_{bma}|P_{raw,m})I_{\{P_{bma}>0\}}
\end{aligned}
\tag{3}
$$

24 where $I_{\{\}}$ is unity if the condition within the brackets is true and zero otherwise. $f(P_{bma}=0|P_{raw,m})$ is the probability

25 of zero precipitation given by a logistic regression model:

$$f(P_{bma}=0|P_{raw,m}) = \frac{1}{1 + exp(b_0 + b_1 P_{raw,m}^{1/3} + b_2 \delta_m)} \tag{4}$$





where $b_0$, $b_1$ and $b_2$ are regression parameters common for all ensemble members and $\delta_m$ equals 1 if $x_m = 0$ and equal
0 otherwise.
$h(P_{bma}|P_{raw,m})$ was assumed to follow a gamma distribution for the cube root transformation $P'_{bma} = P_{bma}^{1/3}$ of the
precipitation, where the mean ($\mu_m$) and variance ($\sigma_m^2$) of the distribution depend on the ensemble member:

$$\mu_m = c_0 + c_1 P_{raw,m}^{1/3} \text{ and } \sigma_m^2 = d_0 + d_1 P_{raw,m} \tag{5}$$

where all parameters $c_0$ and $c_1$ , $d_0$ and $d_1$ were the same for all ensemble members. The seven parameters in the
Bernoulli-gamma kernels were estimated using the catchment average precipitation from seNorge as reference.
**3.3.3    BMA for streamflow (Qbma)**
We applied a Box-Cox transformation (Box and Cox 1964; e.g., Duan et al. 2007) on both observed and forecasted
streamflow to make the transformed streamflow $q^*$ normally distributed:

$$q^* = \begin{cases} \dfrac{(q^\lambda - 1)}{\lambda} \ for \ \lambda \neq 0 \\ log(q) \ \ for \ \lambda = 0 \end{cases} \tag{6}$$

here $\lambda$ is a transformation parameter. The Box-Cox transformation has proven valuable for hydrological applications
(e.g., Engeland et al. 2010; Bates and Campbell 2001; Thyer et al. 2002; Yang et al 2007). We used a fixed $\lambda$ based on
previous studies by Engeland et al (2010), who found that $\lambda$ = 0.2 gave forecast errors that were approximately
independent of forecasted values. As for temperature, we applied the BMA with a mixture of normal kernels for
postprocessing the streamflow forecasts.

$$f\left(Q'_{bma}|Q'_{raw,m}\right) \sim N\left(a_0 + a_1 Q'_{raw,m}, \sigma^2\right) \tag{7}$$

**3.4    BMA training length**
Following Raferty et al. (2005), the BMA models for temperature, precipitation and streamflow were trained on data
from a time window prior to the issue date for each forecast. We tested different training lengths for all variables and
lead times, using CRPS (description in following section) as evaluation metric. Experiments with different training
lengths showed that the optimal window size depends on variable, lead-time, and whether CRPS was calculated for all
data or only for days with flooding (example in Fig 3). Precipitation was most sensitive to the training length due to
the necessity of precipitation occurring within the time window. 45 days training period was optimal for most
catchments and lead-times (A-Fig 1 and 2). To keep a consistency during the evaluation we used 45 days training
period for all variables (i.e., temperature, precipitation, and streamflow).
**3.5    Temperature and precipitation dependence structure (Ensemble copula coupling)**
The BMA models described above were applied independently to each weather variable, each location (here
catchment) and each lead time. The preprocessed ensembles where established by drawing 51 new realizations from





the mixture distribution of each BMA model independently. To recreate forecast trajectories of temperature and
precipitation, it is necessary to account for the temporal and inter-variable dependence structures. In this study, it was
achieved by using an approach similar to Ensemble Copula Coupling (ECC, Schefizik et al., 2013). The original 51
ensemble members ($m$) for temperature and precipitation were, for each location, issue date, and lead time, assigned a
rank ($r_{o,m}$). Similarly, the 51 BMA-processed precipitation and temperature ensemble members were assigned a rank
($r_{n,m}$). The 51 preprocessed ensemble members were reordered by using $r_{o,m}$ and $r_{n,m}$ as keys to keep the preprocessed
ensemble in the same rank sequence as the original ensemble members. By applying this method to all variables, lead
times, and issue dates we maintain the dependency between the variables, as well as the temporal dependency for each
of the variables.
**4    Evaluation**
We evaluated the pre- and postprocessing methods for all days of the study period using the complete dataset, as well
as for the flood dataset.
**4.1    Reliability: Cumulative rank-histogram plots**
The reliability of an ensemble forecast is often visually presented by the rank-histograms (Anderson, 1996; Talagrand
et al., 1997; Hamill, 2001). In our setup, the rank-histograms consist of $i$=52 bins (51 members +1), where the value
of the ordered ensemble members defines the limit between the bins. Each bin in the rank-histogram reflects the
frequency of the ranked reference observations compared to the ensemble forecast, and a reliable forecast should have
a uniform distribution of observations between the bins. There are 14 rank-histograms for each lead time and catchment
to be evaluated. To reduce the number of plots, we evaluated the reliability by creating a Q-Q plot based on the
cumulative rank-histogram (scaled to unity) on the y-axis and the uniform distribution on the x-axis, as explained in
Fig 4. The cumulative rank-histogram $F_i$ for bin $i$ is the sum of the relative frequency $f_k$ for all bins where $F_i =$
$\sum_{k=1}^{i} f_k$. The expected relative frequency of observations in each of the 52 bins given a uniform distribution equals
1/52, represented by the cumulative uniform distribution $U_i = \sum_{k=1}^{i} \frac{k}{52}$. . In this cumulative rank-histogram plot the
1:1 line represents a uniform rank-histogram with an equal probability for the observations to be located within each
bin. This approach enabled us to compare the reliability for all 14 processing schemes within a single plot. The shape
of the cumulative rank histogram plots enables the detection of biases as well as under- and over dispersion as explained
in the Fig 4.
**4.2    Continuous rank probability score (CRPS) and - skill score (CRPSS)**
The continuous rank probability score (CRPS) has properties that are appealing for the evaluation of ensemble forecast.
Firstly, it is sensitive to the entire permissible range of parameters of interest. Secondly, its definition does not require
predefined classes, which might influence the results. For a deterministic forecast, CRPS reduces to the mean absolute
error (MAE, Hersbach, 2000), which enables a comparison between a deterministic and an ensemble forecast. CRPS
measures the integral of squared difference between the forecast and the observation, both given as cumulative
distribution function (cdf). If the observation is deterministic the Heaviside function is used for the observation cdf
(Hersbach, 2000). For ensemble forecasts, the CRPS is calculated discretely since both the observations and the
forecasts are reported in discrete intervals (Hersbach, 2000, Eq. 8):





$$CRPS = \frac{1}{M} \sum_{m=1}^{M} |x_m - x_{obs}| - \frac{1}{M^2} \sum_{m=1}^{M} \sum_{n=1}^{M} |x_m - x_n| \tag{8}$$

Where $M$ is the ensemble size, $x_m$ is ensemble member $n$ and $x_{obs}$ is the reference observation. For a time-series of
forecasts, the mean CRPS for each scheme ($\overline{CRPS_{PS}}$) can be calculated. CRPS will give credit to high probabilities
close to the reference, which is not necessarily the case for other ensemble verification scores (Gneiting and Rafterty,
2007). CRPS has the same unit as the observations ($m^3$/s for streamflow), and is negatively oriented, where zero is the
optimal value.
The continuous ranked probability skill score (*CRPSS,* Eq. 9) enables assessment of the skill of the different processing
schemes (*PS*) relatively to the raw forecasts (raw). The mean CRPS for each scheme ($\overline{CRPS_{PS}}$) and for the unprocessed
forecasts ($\overline{CRPS_{raw}}$) are used to calculate CRPSS.

$$CRPSS_{PS} = 1 - \frac{\overline{CRPS_{PS}}}{\overline{CRPS_{raw}}} \tag{9}$$

Note that CRPSS has 1 as the optimal value and is positively oriented. Since CRPSS has no units, we could calculate
average skill scores across all catchments. *CRPS* and *CRPSS* were calculated for the complete dataset as well as well
as for the flood dataset.
**4.3    The Critical success index (CSI)**
In an operational flood forecasting setting, flood warnings are issued when there is a certain probability for streamflow
to exceed predefined flood warnings thresholds. The occurrence and non-occurrence of floods are therefore binary
events that can be summarized in a contingency table providing an overview of hits (H), missed events (M), false
alarms (F), and correct non-events (N). Based on the contingency table shown in Table 3, the following indices can be
used to evaluate the performance of a forecasting system.
Hit ratio, where a hit rate of 1 is the best performance (S$_R$):        $S_R = \frac{H}{H+M}$
False alarm ratio (F$_R$):        $F_R = \frac{F}{H+F}$
Critical Success Index (CSI):        $CSI = \frac{H}{H+F+M}$
Since floods are rare events, there is a small number of flood-events compared to the number of non-events. A good
forecast has a high hit ratio and a low false alarm ratio. The Critical Success Index (CSI, Donaldson et al., 1975; Jolliffe
and Stephenson, 2018) balance these two aims by penalizing the hit ratio for both the missed events (M) and the false
alarms (F). In an operational setting, a warning will be issued when a predefined number of ensemble members (or a
defined probability) exceeds the flood warning threshold. The probability of exceedance opens for potential cost lost
evaluation, however for the simplicity of this work we have chosen a limit of 10 members exceeding the mean annual
flood level. The mean annual flood has of a return period of 2.33 years (i.e. ~20% probability of occurrence).
**4.4    Floods by seasons**





There might be several reasons for the seasonal differences in flood forecast performance. Firstly, there are biases in
forecasted temperatures, especially for the Norwegian coast during autumn and winter (Seierstad et al., 2016, Hegdahl
et al., 2019). Secondly, the flood-dominating processes are often aligned to different season, e.g. snowmelt contribution
to floods dominates in spring, and rain-induced floods dominate in autumn. For these reasons, we divided the flood
events into spring and autumn floods and used *CSI* to evaluate how the performance of processing methods depend on
season. The available data covers a period of two years and we defined spring from April 4 to June 13, and autumn
from September 01 to December 10. Both seasons consist of $2 \times 101$ days and 35 catchments were affected by spring
floods and 40 catchments by autumn floods.
**5    Results**
We assessed the reliability of the raw and processed streamflow forecasts, and results for selected catchments are
presented. *CRPS* and *CRPSS* were used to evaluate the different processing schemes for the full dataset and the flood
dataset. Furthermore, we evaluated the effect of pre- and postprocessing regarding location, by plotting maps of the
processing schemes giving the highest performance on the flood dataset. *CSI* was used to assess the ability to predict
the exceedance of flood warning levels for the different schemes. *CSI* was calculated for all floods as well as for spring
and autumn floods separately. Finally, we present streamflow forecasts based on the different processing approaches
for three flood events.
**5.1    Reliability**
We used cumulative rank-histogram plots to compare all 14 processing schemes for all lead times and found that for
most catchments the schemes improved the reliability of the forecasts. Examples for lead times 1, 5 and 9 for three
catchments chosen to highlight some differences, are shown in Fig 5. Vaekkava (Fig 1, Table 1) is representative of
the effect of pre- and postprocessing for most catchments in this study. The raw ensembles (*Traw_Praw*) have a
negative bias for all lead times. For a lead time of 1 day, all postprocessing schemes produce reliable forecasts, whereas
preprocessed forecasts still underestimate the streamflow forecasts. The preprocessed forecasts become more reliable
with increasing lead time. This can be explained by an increasing spread in the ensemble for longer lead times. For a
lead time of 9 days, we see that the preprocessed forecasts, independent of methods, are more reliable than the
postprocessed forecasts. Refsvatn (Fig 5 second row, Table 1) has a slightly positive bias in the raw ensemble
(*Traw_Praw*) for a lead time of 1 day. The preprocessing schemes results in forecasts with a large negative bias and
hence makes the forecasts less reliable, whereas schemes with postprocessing (*\*_Qbma*) improve the reliability. For
lead times of 5 and 9 days, the raw ensembles are the most reliable. Tannsvatn (Fig 5 bottom row, Fig 1, Table 1) has
raw forecasts that are rather reliable for all lead times. For a lead time of 1 day, the improvements are seen by all
postprocessed ensembles whereas the preprocessing introduces a negative bias. For a lead time of 5 days, the reliability
is similar for most processing schemes, but poorest for the preprocessing schemes *Pcal* and the *Tbma*. At a lead time
of 9 days, however, the preprocessing schemes based on *Pbma*, performs best, while those that include postprocessing
are least reliable.
**5.2    Skill – relations to lead time for all data and floods**





We used CRPS and CRPSS to evaluate how the different processing methods affected the performance of ensemble
streamflow forecasts for all lead times and catchments. In Fig 6 the *CRPSS* for all data and catchments is presented.
The most striking finding is that nearly all catchments benefit from processing. Postprocessing in combination with
preprocessing is most important for the short lead times. *Pcal* show the largest variability in performance, where a
larger portion of catchments only slightly benefit from *Pcal*, indicating that this preprocessing is the least robust. For
the flood dataset (Fig 7), there is a larger difference between the median of *CRPSS* for the schemes compared to Fig
6. However, the variability in skill is larger for the flood dataset compared to the full dataset, meaning that there are
fewer catchments benefiting from the processing schemes under flood conditions. Postprocessing without
preprocessing seems to be the least good approach. For the longer lead times, there are increasingly more catchments
where postprocessing leads to a poorer performance, compared to using the raw forecast.
Additional results are shown in A-Fig 3 and 4 for the full dataset and A-Fig 5 and 6 for the flood dataset, all these
figures are in the appendix. By only focusing on the best processing approach for the single catchments, the applied
postprocessing methods are most important for the short lead times (1-3 days) when analyzing the complete dataset
(seen by the yellow to green colors in A-Fig 3 and supported by the histograms in A-Fig 4). We moreover find that the
most skillful method can change for a catchment with lead-time (A-Fig 3). The BMA applied to temperature and in
the combination of BMA applied to precipitation are the two best methods for lead-times above 3 days.
For the flood dataset we find that there are no systematic patterns to whether pre- or postprocessing is most important
to improve the skill (A-Fig 5). Postprocessing performs similar to preprocessing for most lead-times and is hence less
important for the short lead-times compared to what was found for the full dataset. BMA seems to be the better choice
for preprocessing, and improves the performance for more catchments compared to CAL. For longer lead times, BMA
on temperature is the most important method for improved *CRPS*. The general tendency seems to be that preprocessing
precipitation is most important for the short lead-times, whereas preprocessing temperature is more important for the
longer lead-times.
Figure 8 gives a detailed presentation on how the mean *CRPS* varies with lead time, processing scheme, and the
evaluation dataset for three individual catchments. For the full dataset (Fig 8 left), the *CRPS* for postprocessed forecasts
increases faster with lead time than *CRPS* for forecasts without postprocessing. The lead time at which postprocessing
gives better performance than not using postprocessing varies between catchments. This is supported by the results
presented in A-Fig 3. A striking difference is that *CRPS* increases with lead time when the full dataset is used, whereas
it is reduced by lead time for the flood dataset (Fig 8 right) for several of the processing schemes. The pattern for the
full dataset (i.e. *CRPS* increases with lead time) is representative for most catchments, whereas changes in *CRPS* with
lead time for the flood dataset varies between the catchments.  We see that the mean *CRPS* for all streamflows (Fig 8
left) is smaller than for floods (Fig 8 right), which can be explained by the data used to estimate the mean. The flood
dataset consists of fewer days and higher values, and hence the possibility for larger errors. An explanation for the
decrease in *CRPS* for the flood dataset in Fig 8 right is that the ensemble spread increases with lead time, and it is
therefore more likely that the observed floods are within the ensemble range for the long lead times.
**5.3    Skill – relations to location**



Figure 9 shows a map of which processing method that achieves the highest performance according to *CRPS* for the
flood dataset for each catchment for lead time 1, 5, and 9 days. The left column shows whether a preprocessing scheme
alone or a combination of pre- and postprocessing methods gives the highest performance. The figures show that
inland, high elevation, and eastern catchments are improved by postprocessing for lead times of 1 and 5 days, whereas
the coastal catchments do not attain the highest score by postprocessing. In the right column we show which of the
BMA preprocessing approaches that resulted in the best *CRPS*. Catchments where the grid-calibration or the raw
forecasts gave the best performance are shown as black dots. We find that *Pbma*, alone or in combination with *Tbma*,
gives the best results for western and southern coast of Norway for lead times of 1 and 5 days. For a lead time of 9
days, however, *Tbma* alone is more important. In the coastal regions, floods are mainly rain driven, and we find that
*Pbma* performs well in these regions. BMA on temperature alone has a less clear pattern. A summary of the numbers
from Fig 9 is presented in table 4 and quantifies the visual information from Fig 9. The effect of postprocessing is
larger for shorter lead time and the catchments where preprocessing was the best option, BMA is the best choice for
about 70 to 80 % of the catchments. Combining *Tbma* and *Pbma* performs best for a larger group of catchments.
**5.4    CSI for the whole year, spring, and autumn floods**
In this evaluation, the processing scheme giving the highest CSI for each catchment is considered, and we counted the
number of catchments for which the specific scheme gave the best CSI. For each catchment, multiple methods can
achieve equal CSI. Therefore, for some lead times, the number of "best" CSI exceeds the total number of catchments.
We first evaluated CSI for floods from the whole year (A-Fig 8), which did not give any clear indications of methods
that performed better than others. However, by separating the flood dataset between floods occurring in spring (Fig.
10) and those occurring in autumn (Fig 11) we attain some interesting insight. For spring (Fig 10) most methods give
good results for multiple catchments, indicating more than one successful method. The improved predictions by
applying pre- and/or postprocessing to spring floods, holds for most lead times. For lead times of 2 to 5 days
postprocessing provides the best CSI for more catchments than preprocessing alone, whereas beyond 5 days' lead time
we find that about half of the successful predictions includes postprocessing.
For autumn (Fig 11) the results diverge from the spring results. For a lead time of 1 day, the predictions are highly
improved by including postprocessing, whereas the effect of postprocessing diminish for lead times of 2 and 3 days.
From a lead time of 4 days there is no predictability by most methods, and only six catchments show predictive skill
by applying *Tbma* alone or in combination with *Pbma*.
**5.5    The effect of pre- and postprocessing for a selection of events and catchments**
The forecasted streamflow is essential to determine a correct flood warning level. In this subsection we present three
flood events and catchments to exemplify how the different processing approaches influences the ensemble flood
forecasts. The events are the atmospheric river affecting western Norway in October 2014, the extreme weather event
*Synne* hitting southern Norway in early December 2015, and a snowmelt flood in eastern Norway in May 2014. For
all examples, the issue date of the forecast is selected 3 to 5 days before the peak of the flood.
Figure 12 shows the outcome of the different processing approaches for the October 2014 event at Bulken (Fig 1, Table
1) in western Norway. Some of the ensemble members reach the reference streamflow (black line) when *Pbma* is





applied without *Qbma*. However, none of the ensemble medians reach up to the level of the reference streamflow
(black line). *Pbma* induces very high streamflow for some of the members, whereas BMA applied to streamflow
removes the effect of *Pbma* (Fig 12 left and right respectively). The large spread in streamflow when using *Pbma*
indicates large uncertainty in the precipitation forecasts for this event.
The extreme weather event in December 2015 was difficult to forecast. In particular, the location of the rainfall was
highly uncertain. Figure 13 shows the outcome of the different processing approaches for this event at Moeska (Fig 1,
Table 1) in south-western Norway. We see that precipitation is underestimated, and none of the processing schemes
result in ensemble members that reach the reference level for streamflow. For this event at Moeska the same pattern is
seen as for the event at Bulken, where *Pbma* induces high streamflow values (Fig 13 left) that are later suppressed by
the *Qbma* (Fig 13 right).
Figure 14 shows the outcome of the different processing approaches for the snowmelt flood in May 2014 at
Nybergsund in eastern Norway. This flood is best forecasted by the raw and preprocessed input, with small differences
between the schemes. Postprocessing reduces the median forecasts for all lead times, in addition to increasing the
spread.
**6  Discussion**
The results demonstrate that all catchments benefitted from one or more of the applied processing schemes, thereby
confirming our working hypothesis. However, it was not possible to identify a distinct processing chain that was
optimal for all forecasts, the choice of method depends on several factors including lead time, season, location, and
evaluation criteria.
A part of the answer to our first research question "Which pre- and postprocessing approaches should be used in the
hydrometeorological forecasting chain to improve streamflow forecasts with emphasis for flood forecasting?" is that
preprocessing using catchment specific BMA generally performed better than the gridded calibration (CAL). One
explanation is that the BMA calibration uses the same temperature and precipitation data that were used to tune the
hydrological models and establish the reference streamflow. Using grid-calibrated temperature and precipitation might
therefore, in many cases, lead to biases in streamflow forecasts. One example is Refsvatn (Fig 5 LT: 1) where the CAL
methods induce a larger bias compared to the BMA methods. Another aspect is that the BMA approaches tailor the
preprocessing to each catchment, whereas the model for the grid calibrated precipitation is independent of location and
is therefore less flexible (See Table 2). In Fig 6 we see a large variability in performance for *Pcal*. Even though *Pcal*
performs well for a majority of the catchment when considering the full dataset, several catchments show only small
or no improvement to the forecast skill. Postprocessing, i.e. combining *Pcal* and *Qbma*, assists in improving the
forecasts for these catchments.
It is moreover instructive to see that postprocessing alone seems to be the least optimal choice when evaluating both
the full dataset and even less optimal when the subset of floods is considered. This demonstrates the importance of
correcting biases and spread in the forcing. The catchments' responses to the temperature and precipitation inputs are
non-linear, in particular for snow accumulation and snow melt processes where temperature thresholds are important.
Using postprocessing alone is therefore less effective in correcting for biases in inputs to the hydrological model.





The combination of pre- and postprocessing approaches that outperforms the others depends on catchment, lead time,
streamflow magnitude, and the choice of evaluation metric. We find that for the complete dataset, the best CRPS is
seen when applying postprocessing combined with BMA preprocessing of temperature for lead times of up to three
days, whereas for the longer lead times BMA preprocessing of temperature alone or both precipitation and temperature
provide the best performance (Fig 6 and A-Fig 4). This result is in line with Benninga et al (2017) who underlines the
importance of improving the meteorological inputs, in particular for high flow events. Global meteorological
ensembles often lack spread for shorter lead times since they are designed for medium range forecasts and therefore
use perturbations that optimize the ensemble spread for longer lead times. BMA models used both for pre- and
postprocessing will therefor improve the forecast skill. It would be instructive to assess whether using regional
meteorological ensembles, which are better able to model the forecasts uncertainties in the short range compared to
their global counterparts (Frogner et al 2019a, 2019b), as inputs to the hydrological model alter this finding. However,
such forecasts were not available for our study period, but may be the focus of future research.
Comparing CRPSS in Fig 6 and 7, we see that the improvement in skill resulting from the processing schemes is
smaller for the flood dataset compared to the complete dataset. Looking at CRPS for the full dataset and floods (A-Fig
5 and 6 respectively) it is less evident whether any schemes outperform others for the floods whereas for full dataset
we see similar results as for CRPSS. We see that postprocessing is less useful for the three first lead times for the flood
dataset as compared to the full dataset. Using BMA for both precipitation and temperature for the shortest lead times
and only temperature for the longest lead times was the best choice for the largest portion of the catchments. In addition
to the differences in preferred processing schemes between catchments, we find that for a single catchment, the best
processing schemes varies depending on lead-time. This underlines that forecast errors arise from different sources,
and that being conclusive based on relatively small sample of floods is difficult.
In answer to our second research question "*Are there regional or seasonal patterns in preferred pre- and
postprocessing approaches?*" we found that the performance of the processing schemes has both regional and seasonal
patterns, when the flood dataset is used for evaluation. The regional pattern indicates that an excess of catchments
benefitting from preprocessing are located in coastal areas (Fig 9). Another finding is that those improved by BMA
applied to precipitation (*Pbma*) are in areas with high precipitation (the west and southwest coast of Norway, Fig 9).
It is also clear that *Tbma_Pbma* is the combination with the highest performance for a lead time of 1 day,  with the
performance diminishing with lead time, and for a lead time of 9 days, *Tbma_Praw* is a better choice (Table 4 and Fig
9). Postprocessing is more important for the inland and high elevation catchments, where temperature and slower
snowmelt processes are dominating. Moreover, for these regions we see that the effect of postprocessing is smaller
with increasing lead time.
The seasonal effect was evaluated by separating spring floods from autumn floods. The CSI shows that there are large
differences in predictability between seasons. There is almost no ability to predict autumn floods beyond 3 days, only
for 6 of 40 catchments are floods predicted by any of the approaches. In contrast, the forecasts for the spring floods
show a predictability up to 9 days, and for 23 of the 35 catchments one or more approaches were able to predict the
floods. These results indicate that the predictability of floods depends on flood-generating processes, i.e. snowmelt
induced spring floods are easier to forecast than rain induced autumn floods. These results further imply that the autumn
precipitation and floods are the most difficult to predict and has the highest potential for improvements.



For some catchments we see contradictory results when comparing CRPS and CSI for the flood dataset. *Tbma* produces
the best CRPS for most catchments for longer lead-times (A-Fig 6), however *Tbma* gives a lower CSI compared to the
other preprocessing methods (A-Fig 7 and Fig 10-11). This indicates that care must be taken when choosing an
appropriate evaluation metric. CRPS indicates the error between the forecast and the reference value and favors
forecasts close to the reference (Gneiting and Rafterty, 2007). CSI on the other hand gives no favor for forecasts close
to the reference only to whether the forecast exceeds the warning threshold or not. For example, the processing scheme
that had the best CRPS might slightly underestimate the reference value, and if the reference is just above the warning
threshold, this scheme will miss the event, resulting in a low CSI value In contrast, a processing scheme that highly
overestimate the reference will result in a poor CRPS and a good CSI.
For the calculation of CSI, we used a limit of 10 ensemble members (a probability of about 20%) exceeding the flood
threshold to issue a flood warning. The ensemble can provide a whole range of probabilities and here we only evaluated
for one probability level. The optimal probability of exceedance to issue a flood warning might be different between
catchments, lead times, and seasons. Another aspect is to investigate the acceptance level for false alarms to missed
events. The number of tolerable false alarms might depend on the impacts of the event (e.g. risk evaluation), and it is
therefore difficult to make one absolute decision on behalf of all possible exceedance levels (flood sizes) and affected
parties. We acknowledge that the choice of evaluation criteria can be different depending on the users and the cost of
mitigation action compared to the loss due to an event, and that false alarms and missed events might be weighted
different depending on a total cost-loss evaluation.
One concern when using BMA for preprocessing precipitation is that some of the ensemble members in *Pbma* attained
physically non-plausible values. resulting in very high flood forecasts. This is apparent for the Bulken catchment for
the October 2014 event (Fig 12). This suggests that the forecast distribution can be sensitive to large errors in
precipitation. Especially for Western Norway where a steep topography causes large spatial differences in precipitation
and therefore a potential for large errors in forecasts, *Pbma* should be used with care. The region experienced large
amounts of precipitation prior to the October 2014 event. Therefore, the estimated BMA parameters are based on data
for a period with possible large errors in the forecasted precipitation, implicating large uncertainty in the BMA model
parameters. Possible solutions could be to use categorized approached (e.g., Ji et al., 2018), where the precipitation is
separated into precipitation categories (based on for example daily ensemble mean) and unique BMA models are
trained for each category.
**7   Conclusions**
In this study, we have evaluated streamflow forecasts in 119 catchments based on fourteen schemes with different
combinations of the raw, pre-, and postprocessed values. The modelling chain is similar to the operational flood
forecasting system, and we evaluated the forecast with a special emphasis on flood values exceeding the mean annual
flood (QM). From the results presented and discussed in this paper, we conclude that:
Applying pre- or postprocessing schemes improve streamflow forecasts compared to using raw forecasts. The best
combination of pre- and postprocessing approaches depends on location, season, lead time, and the purpose of the
forecasting as represented by different evaluation criterions. The large number of catchments used for evaluation





allows us to draw some general conclusions that can assist us in choosing an appropriate processing chain and to
identify which forecasts that are the most challenging.
*Which pre- and postprocessing approaches should be used in the hydrometeorological forecasting chain to improve*
*streamflow forecasts with emphasis for flood forecasting?*
• An evaluation of CRPS for the complete dataset of two years showed that the combination of pre- and
postprocessing is most effective for short lead times, up to two-three days. For longer lead times, processing
schemes that only include preprocessing provide the best results. BMA is the preferred method for
preprocessing, either applied to temperature (*Tbma*) alone or in combination with precipitation (*Pbma*).
• For days where floods exceeded QM the added value of processing is less clear. For a small majority of the
catchments applying BMA to precipitation and/or temperature (for longer lead times) improves the CRPS
compared to the raw forecast and is also better than grid calibration.
*Are there regional or seasonal patterns in preferred pre- and postprocessing approaches?*
• The processing is sensitive to regional or seasonal patterns. Postprocessing was most effective for inland and
higher elevated catchments. The coastal catchments gained more from preprocessing. Especially BMA
applied to precipitation and temperature improved CRPS for the western and southwestern coastal catchments
for the early lead times, whereas *Tbma* was most important for the longer lead times.
• The added value of processing depends on season. We see a substantial difference between spring and autumn
floods using critical success index (CSI) for evaluation. In autumn, there are almost no predictive skill for
more than 3 days lead-time. Spring is quite different with a longer prediction horizon; for some catchments
and processing schemes the floods are predicted up to nine days in advance. The results indicate a higher
predictability in spring floods, which in addition to precipitation are highly dependent on temperature that
controls the snowmelt intensity.
• The high precipitation rates, which is the flood generating process in autumn, should hence be the focus for
further improvements. We found that for some incidents of high precipitation rates the BMA preprocessing
resulted in unrealistic precipitation amounts for individual ensemble members. Approaches to amend this are
needed.
To summarize; we find that flood forecasts benefit from pre- and/or postprocessing, however the optimal processing
approaches depend on region, catchment, and season.
**8   Acknowledgment**
The authors would like to thank Thomas Nipen and Ivar Seierstad at MET Norway for their aid during the
implementation of https://github.com/metno/gridpp applied for the forecasting setup of this study, and for providing
the parameters used in the grid calibration processing schemes.
**9   Data and scripts**
We have used the R-package ncdf4, ensembleMOS, ensembleBMA, SpecsVerification.



https://github.com/metno/fimex, was used for the resampling and reprojection of the gridded datasets, and
https://github.com/metno/gridpp which includes the preprocessing methods was applied for temperature and
precipitation calibration (CAL).
The SeNorge data are downloadable, https://thredds.met.no/thredds/projects/senorge.html, Met Norway
The ensemble forecast data is available from ECMWF, and streamflow observation is available from NVE upon
request





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





**11    Tables and Figures**
**Table 1 Catchment characteristics for selected catchments: Catchment Area, Annual runoff (Q), Annual precipitation (P),**
**catchment mean elevation (Mean elev), effective lake area (Eff lake), glacier area (Glacier).**

| Name | Area (km$^2$) | Annual Q (mm) | Mean elev (m.a.s.l) | Eff lake (%) | Glacier (%) |
|------|------|------|------|------|------|
| Vaekkava | 2078 | 375 | 414 | 0.87 | 0.00 |
| Refsvatn | 53 | 1843 | 297 | 1.00 | 0.00 |
| Tannsvatn | 118 | 719 | 905 | 4.59 | 0.00 |
| Moeska | 121 | 1585 | 325 | 1.71 | 0.00 |
| Nybergsund | 4425 | 487 | 781 | 2.48 | 0.00 |
| Bulken | 1092 | 2038 | 867 | 0.88 | 0.39 |

**Table 2 Overview of data and parameters applied the different calibration schemes.**

| Variable | Resolution | Reference data | Lead time | Season/ Annual | Training period |
|------|------|------|------|------|------|
| Pcal | Grid ~25km | 200 WMO | - | - | 2014 |
| Tcal | Grid ~25km | Hirlam 5km | Parameters estimated using the first 24 hours, applied to all lead times | Monthly specific parameter values | 2006 to 2011 |
| Pbma | Catchment average | seNorge catchment average | Parameters lead-time specific 1:9 | Parameters specific each issue date | 45 previous days |
| Tbma | Catchment average | seNorge catchment average | Parameters lead-time specific 1:9 | Parameters specific each issue date | 45 previous days |
| Qbma | Catchment average | Sim HBV | Parameters lead-time specific 1:9 | Parameters specific each issue date | 45 previous days |

**Table 3 Contingency table for classification of hits (H), missed events (M), false alarms (F), and correct non-events (N).**

| | | Observation | |
|------|------|------|------|
| | | **No** | **Yes** |
| **Forecast** | **No** | *N* | *M* |



| | Yes | *F* | *H* |
|---|---|---|---|

**Table 4 Summary of the results in Fig 9. ∑Post and ∑Pre shows the number of the catchments where the combination of**
**pre- and postprocessing approaches gave the best performance. % pre shows the percentage of catchments where**
**preprocessing gave the best performance. Tbma_Praw Traw_Pbam, Tbma_Pbma shows which preprocessing scheme**
**using BMA that gave the best performance.**

| | Pre- or postprocessing | | | Preprocessing – BMA | | | | |
|---|---|---|---|---|---|---|---|---|
| Lead time | ∑Post | ∑Pre | % pre | *Tbma_Praw* | *Traw_Pbam* | *Tbma_Pbma* | *∑bma* | *%bma* |
| 1 | 40 | 40 | 50 | 5 | 5 | 20 | 30 | 75 |
| 5 | 37 | 43 | 54 | 12 | 10 | 13 | 35 | 81 |
| 9 | 31 | 49 | 61 | 22 | 6 | 6 | 34 | 69 |

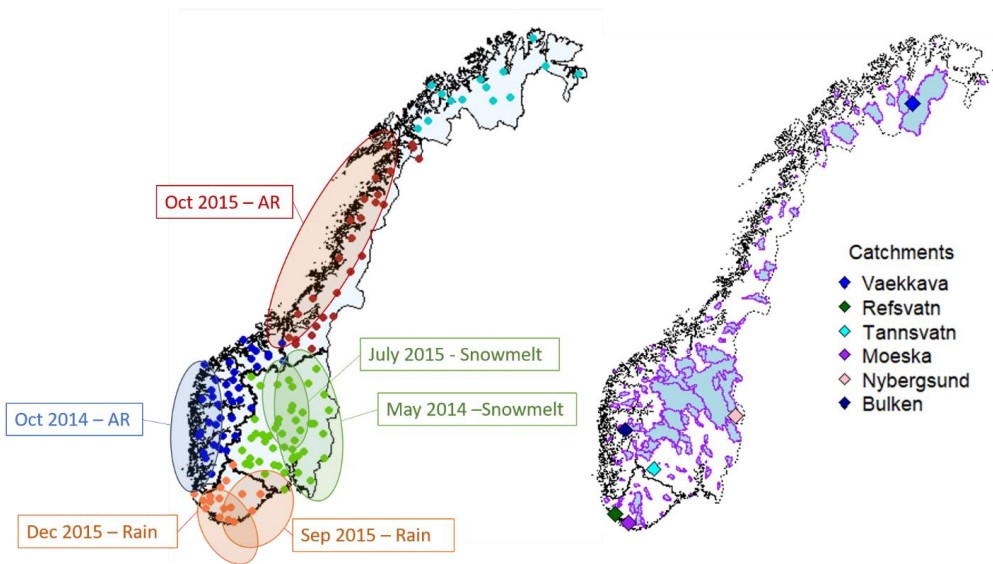

**Figure 1: The map to the left shows the location of the outlet of the 119 catchments used in this study as well as a**
**schematic overview of the areas affected by floods caused by different events (rain, snowmelt and atmospheric river (AR))**
**during the study period 2014 to 2015. It is worth noting that not all catchments experienced floods within the areas. The**
**map to the right shows the catchment areas, and the locations of six catchments for which we will show some detailed**
**results are also shown.**





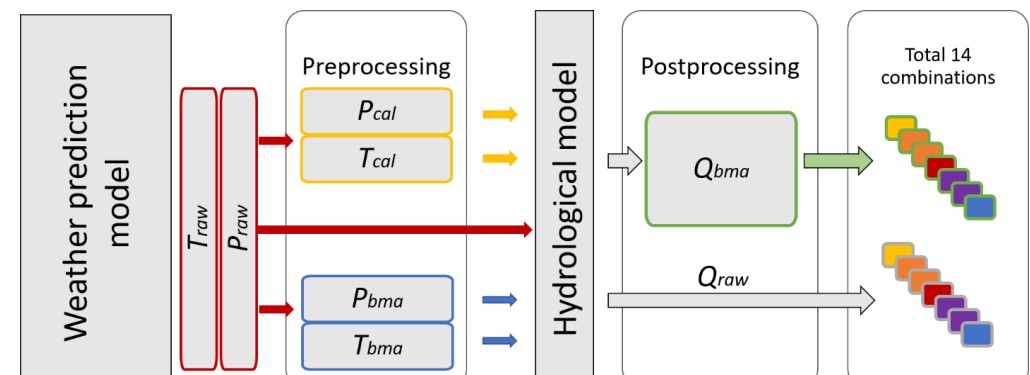

**Figure 2: The processing chain of the experimental set up.** *Traw* and *Praw* are the unprocessed forecasts. Two
preprocessing approaches were applied, a grid calibration (CAL) producing the ensembles *Tcal* and *Pcal*, and Bayesian
model averaging (BMA) producing the ensembles *Tbma* and *Pbma*. All combinations of *Tcal* and *Pcal* together with *Traw*
and *Praw*, as well as all combinations of *Tbma* and *Pbma* together with *Traw* and *Praw*, in total 7 combinations, were run
through the hydrological model. BMA was applied to the streamflow forecasts producing the ensembles *Pbma* in addition
to *Qraw*. In total 14 combinations of pre- and postprocessing were evaluated. The processing schemes were applied to each
issue date, lead time and catchment.

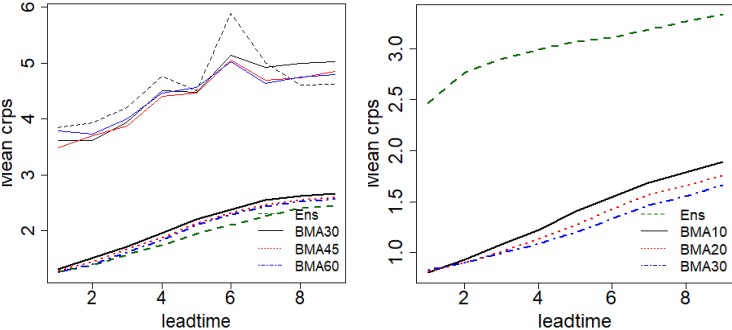

**Figure 3: Left: Precipitation mean CRPS for all lead times for the Aulestad catchment. Thin lines are the 10% percentile
precipitation, thicker lines include all the data. Right: temperature mean CRPS for all lead times for Viksvatn.**



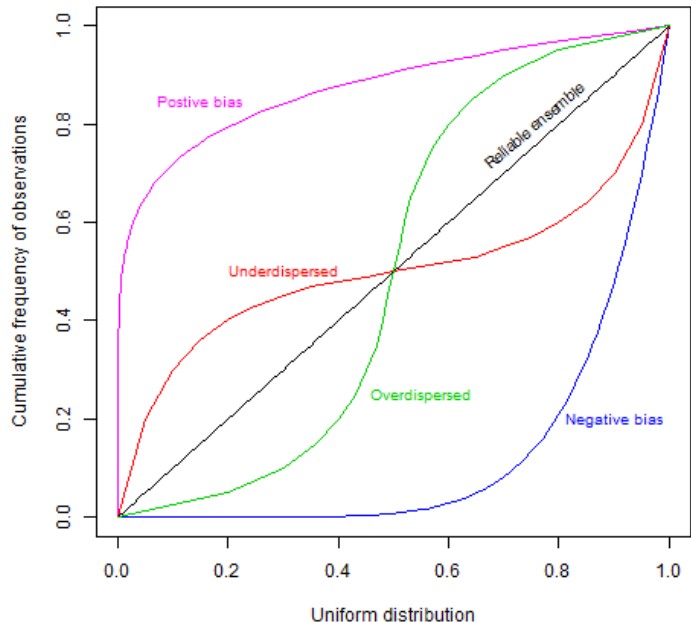

**Figure 4: Typical shapes of the cumulative rank-histogram plots that can be used to detect both biased, over- and**

**underdispersed ensembles. The closer the curves are to the 1:1 line, the more reliable are the ensembles.**



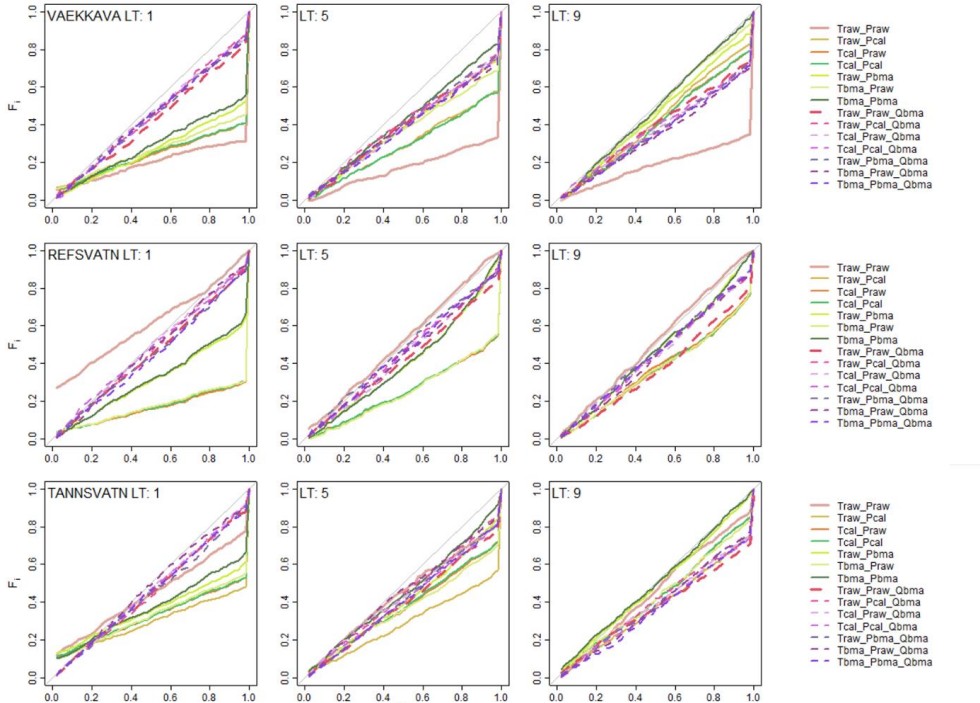

**Figure 5: Reliability plots that compare all 14 processing schemes for Lead time 1, 5 and 9 days (LT: 1,5,9))for three**
**catchments. The location of the catchments is shown in Fig 1 right. The cumulative empirical rank-histograms scaled to**
**unity is shown on the y-axis whereas the uniform distribution is shown on the x-axis. The most reliable forecasts are**
**closest to the 1:1 line. Fig. 4. provides details for interpretation of these plots.**



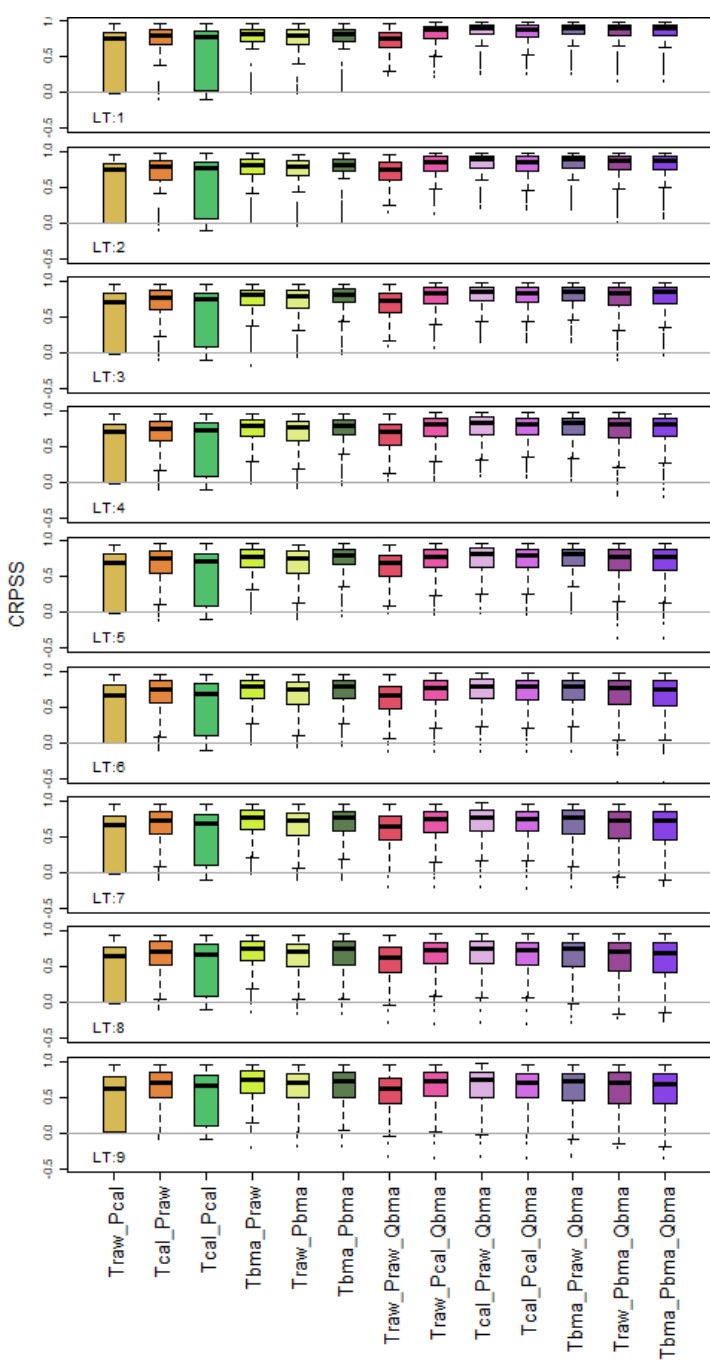

**Figure 6: Boxplot of CRPSS (best is 1) for all catchment based on the full dataset for all processing schemes (x-axis) and**

**all lead times (rows). The first six boxplots indicate the different preprocessing schemes, whereas the last seven indicates**

**processing schemes that includes a postprocessing step.**





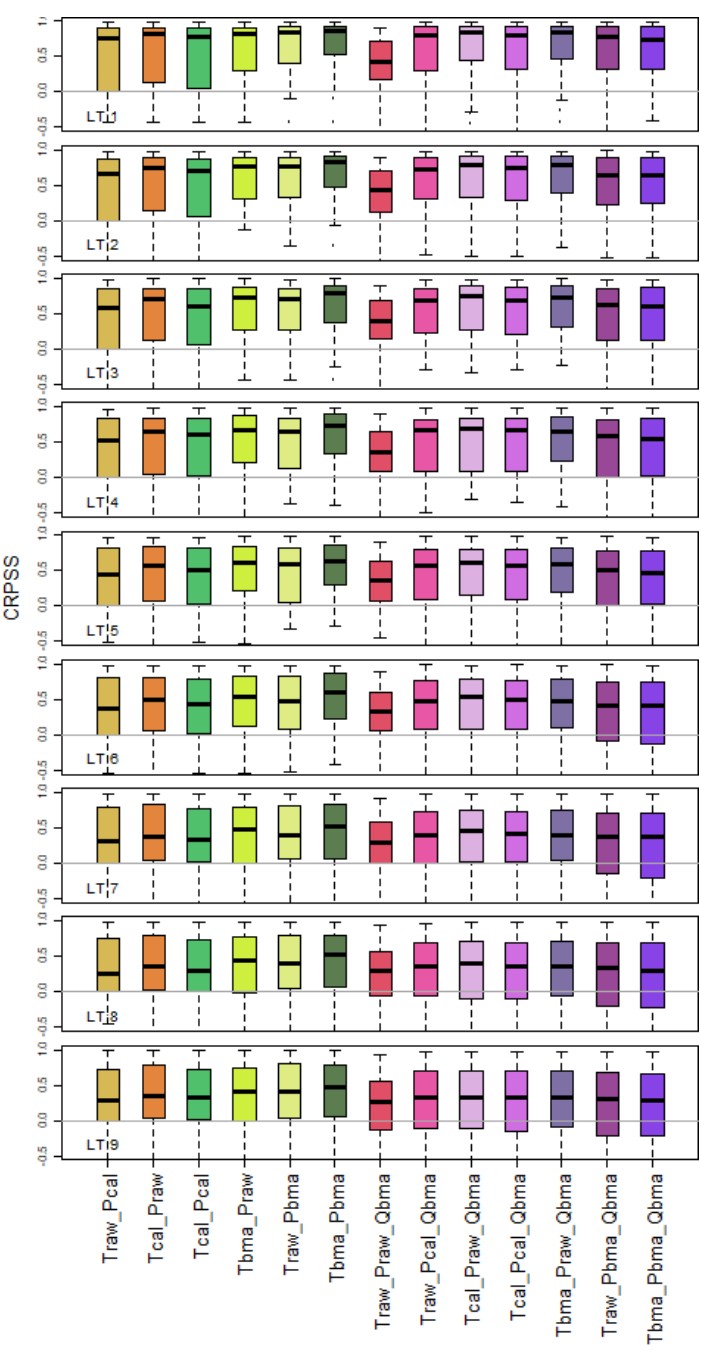

**Figure 7: Boxplot of CRPSS (best is 1) for all catchment based on the flood event dataset for all processing schemes (x-**

**axis) and all lead times (rows). The first six boxplots indicate the different preprocessing schemes, whereas the last seven**

**indicates processing schemes that includes a postprocessing step.**

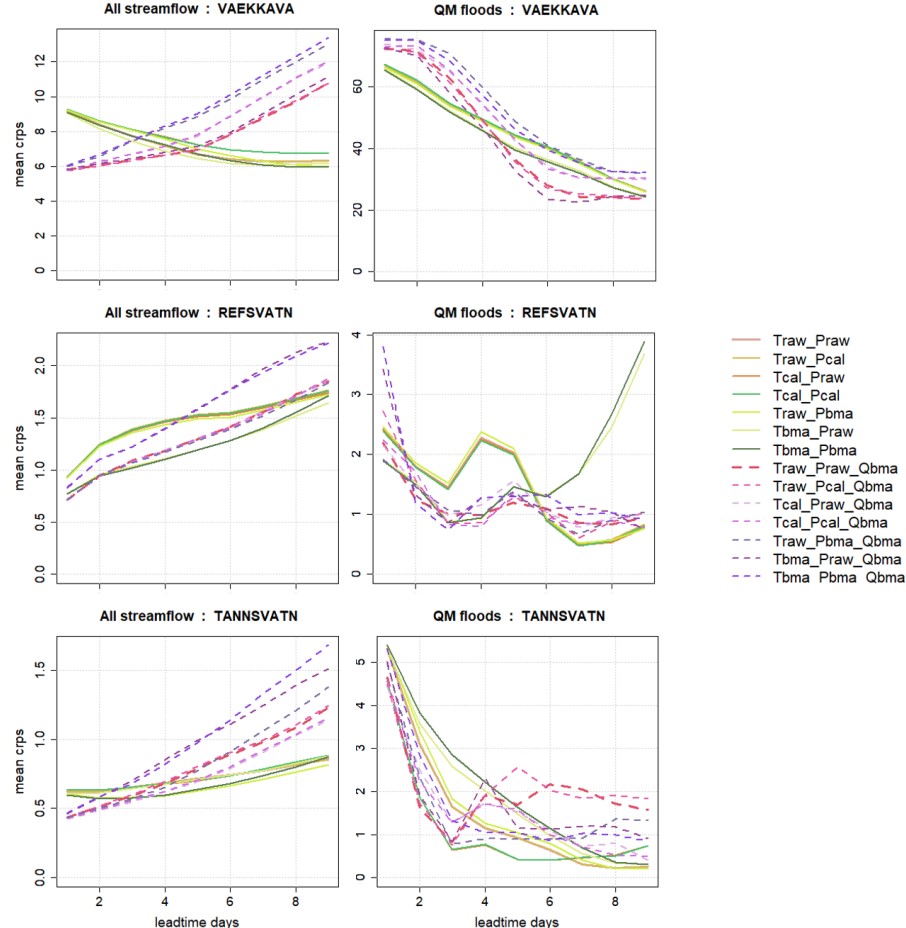

**Figure 8: Mean CRPS (in m³s⁻¹) for three selected catchments as a function of lead time calculated for the full dataset to**
**the left, and the flood dataset to the right. Note that the values for "mean CRPS" on the y-axis is different for the different**
**plots. For Vaekkava 13 days used to calculate the flood dataset (May 30 - June 5 2014 and May 24-30 2015), whereas 2**
**days were used for Refsvatn (December 05-06 2015) and Tannsvatn (May 21-22 2014).**

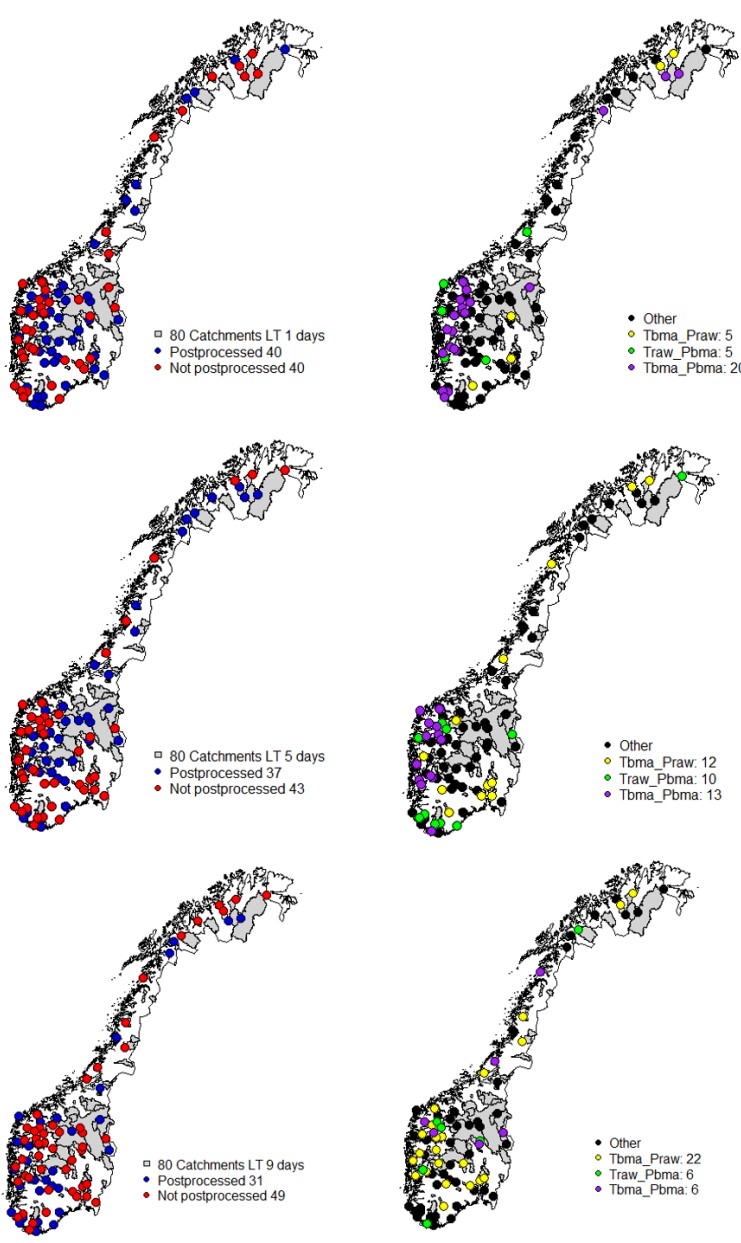

**Figure 9: Figures to the left indicates catchments where any preprocessing approaches alone (red dots) or the combination**
**of pre- and postprocessing (blue dots) provides the highest performance evaluated by the mean CRPS for lead times of 1,**
**5, and 9 days. The figures to the right show the BMA preprocessing scheme that provides the best CRPS. All evaluation of**
**CRPS was applied for the subset of floods.**



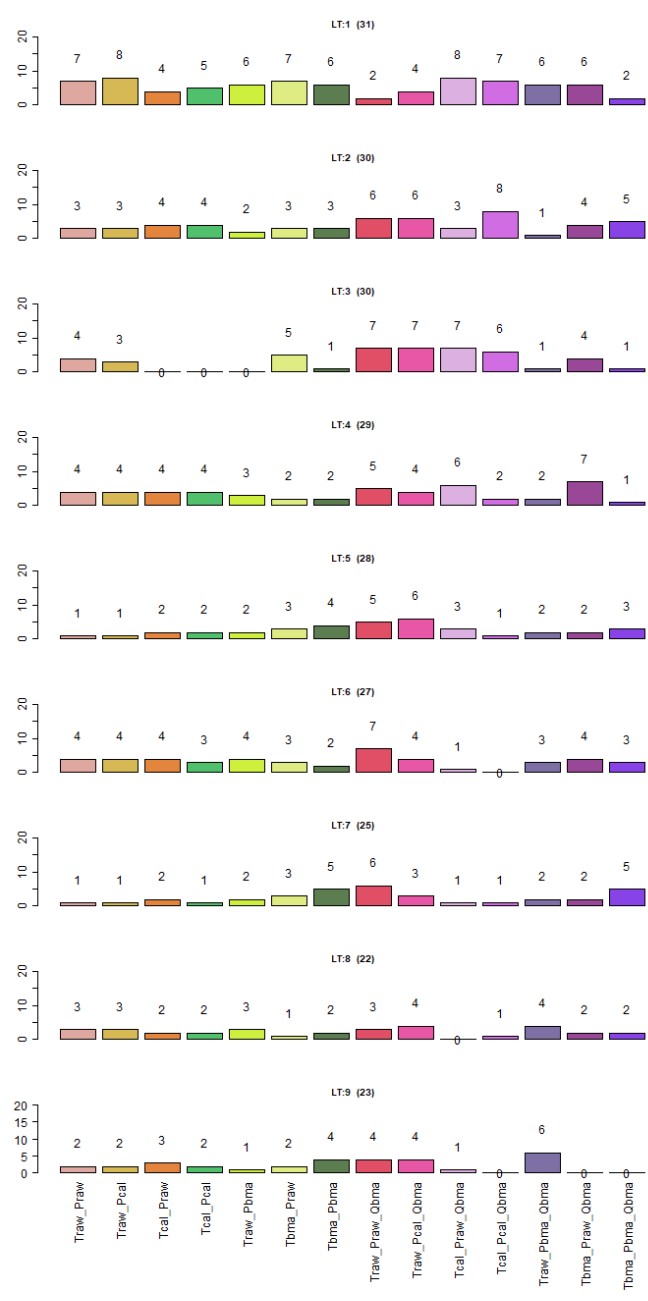

**Figure 10:** Spring**- Critical success index (CSI). Each row represents one lead time (from 1 to 9 days) and includes all processing schemes. In parenthesis the total number of catchments that predicted the exceedance of warning level.**

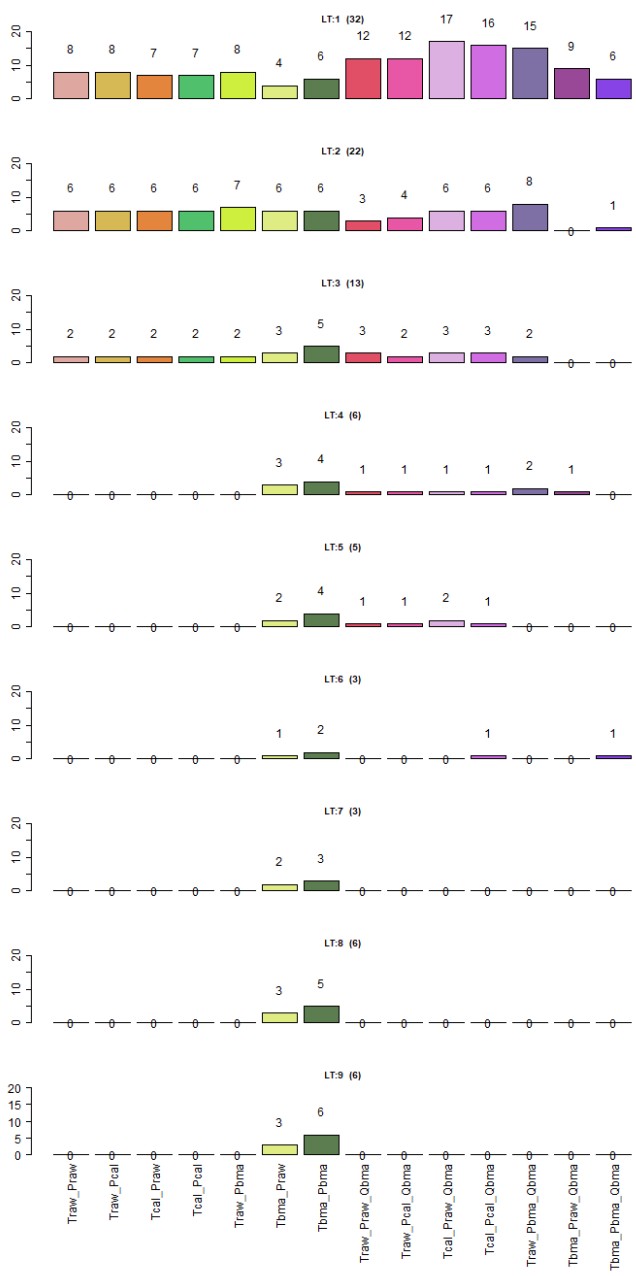

**Figure 11: Autumn- Critical success index (CSI). Each row of barplots represent one lead time (from 1 to 9 days) and includes all processing schemes. In parenthesis the total number of catchments that predicted the exceedance of warning level.**

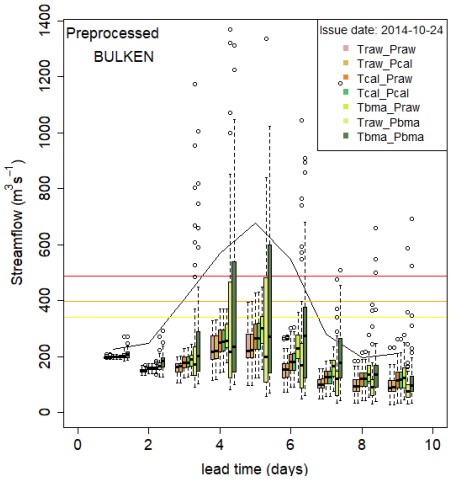
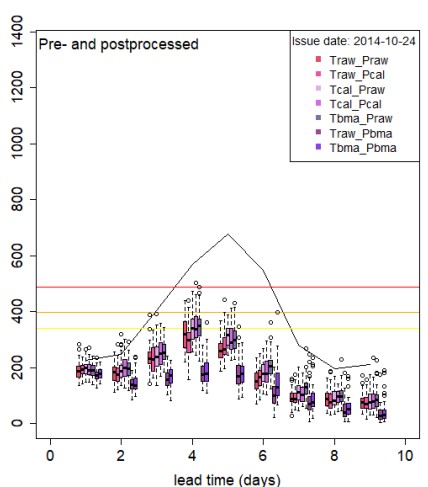

**Figure 12: The AR event 2014 at Bulken. Boxplots of the applied processing schemes. The black line indicates the**

**reference streamflow for the event. The horizontal lines represent the mean annual flood (yellow), the 5-year flood**

**(orange)and the 50-year flood (red).**

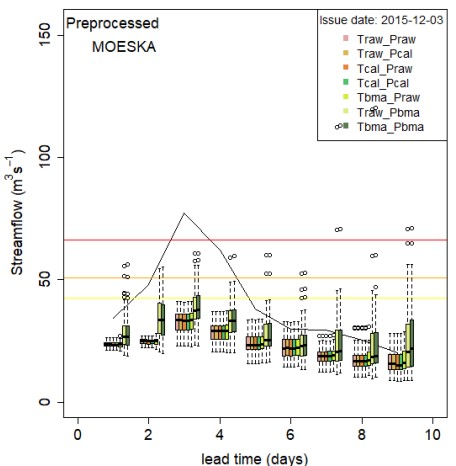
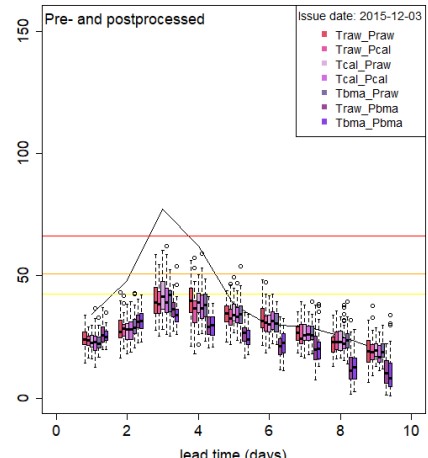

**Figure 13:The extreme weather event Synne in 2015 at Moeska with boxplots indicating the streamflow estimates for**

**different processing approaches. Reference streamflow for the event is the black line. The horizontal lines represent the**

**mean annual flood (yellow), the 5-year flood (orange)and the 50-year flood (red).**



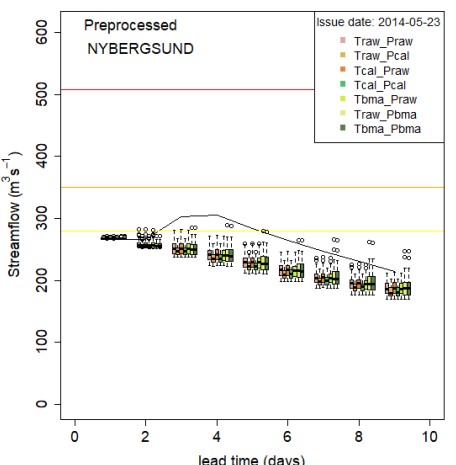
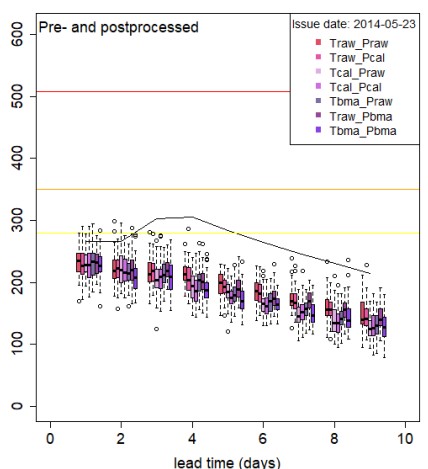

**Figure 14: The snowmelt flood in May 2014 at Nybergsund, with boxplots indicating the streamflow estimates for different processing approaches. Reference streamflow for the event is the black line. The horizontal lines represent the mean annual flood (yellow), the 5-year flood (orange)and the 50-year flood (red).**





## 11.1 Appendix-Figures

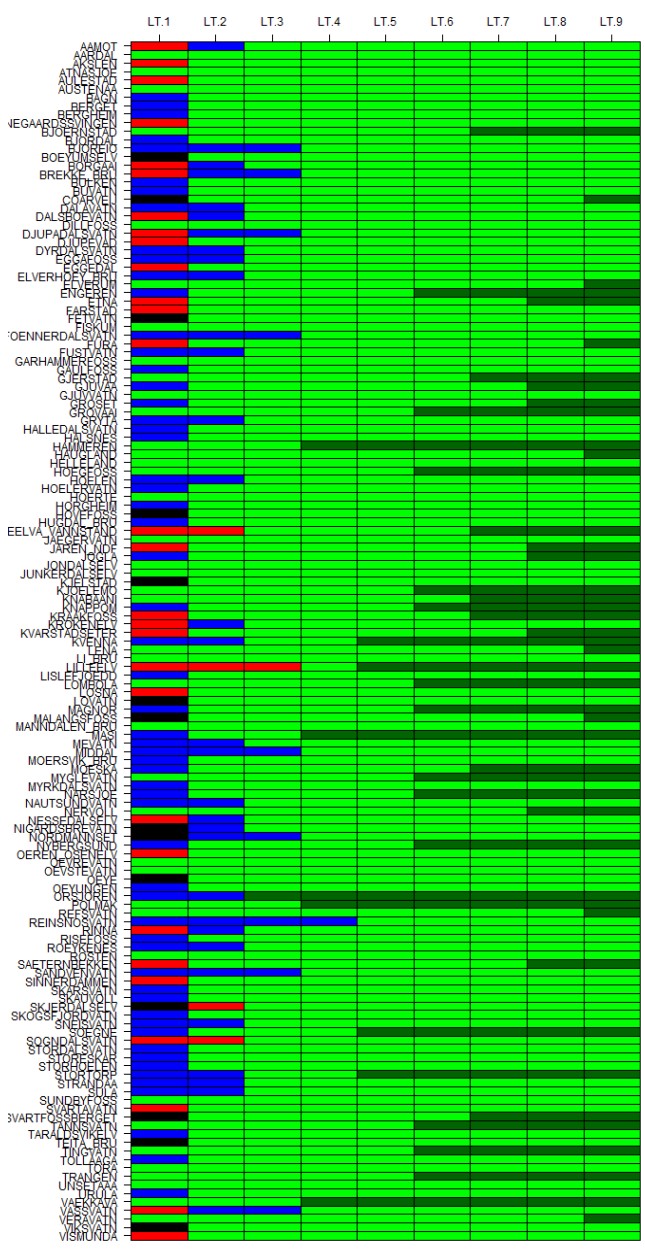

A-Figure 1: Optimal training length for temperature forecasts, using CRPS as evaluation criterion for all catchments (rows) and all lead-times (columns). Table indicates all catchments (rows) and lead-times (columns). Black: 10 days, red: 20 days, blue: 30 days, green: 45 days, dark green: raw ensemble.

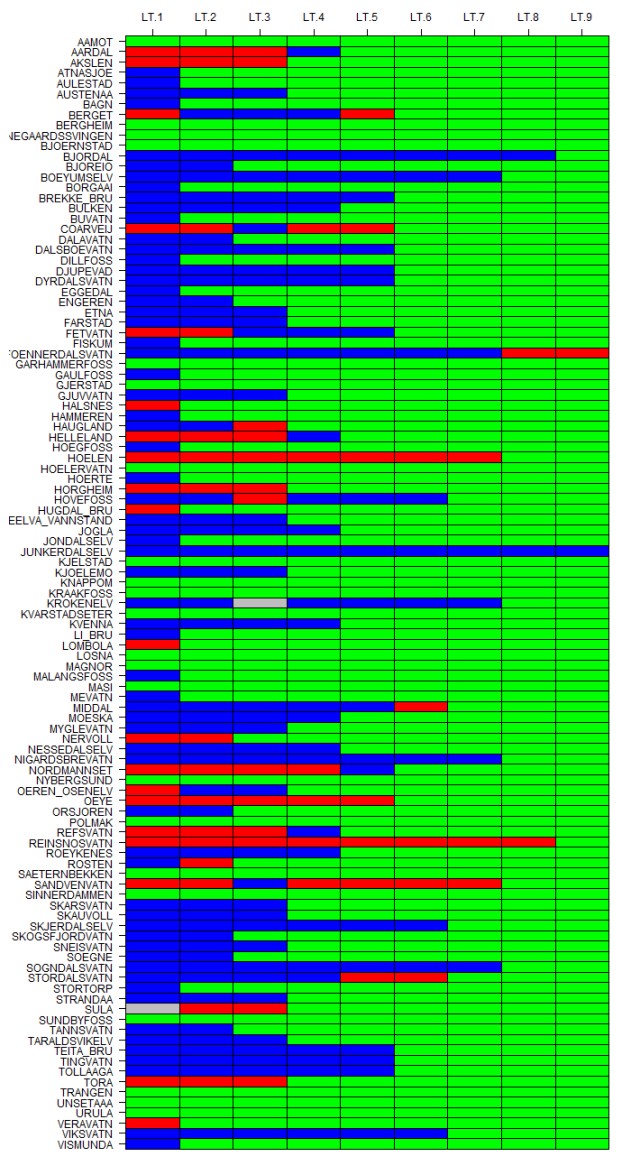

**A-Figure 2: Optimal training length for precipitation forecasts, using CRPS as evaluation criterion for all catchments**
**(rows) and lead-times (columns). Red: 20 days, blue: 30 days, green: 45 days. BMA applied to precipitation depends on**
**sufficient number of precipitation values above 0 to converge. We found that for some catchments this was a problem, and**
**this was also important for the decision to use a 45 days training window, even though the results from the figure shows**
**that for some catchments and lead times, the CRPS is better for shorter training lengths.**



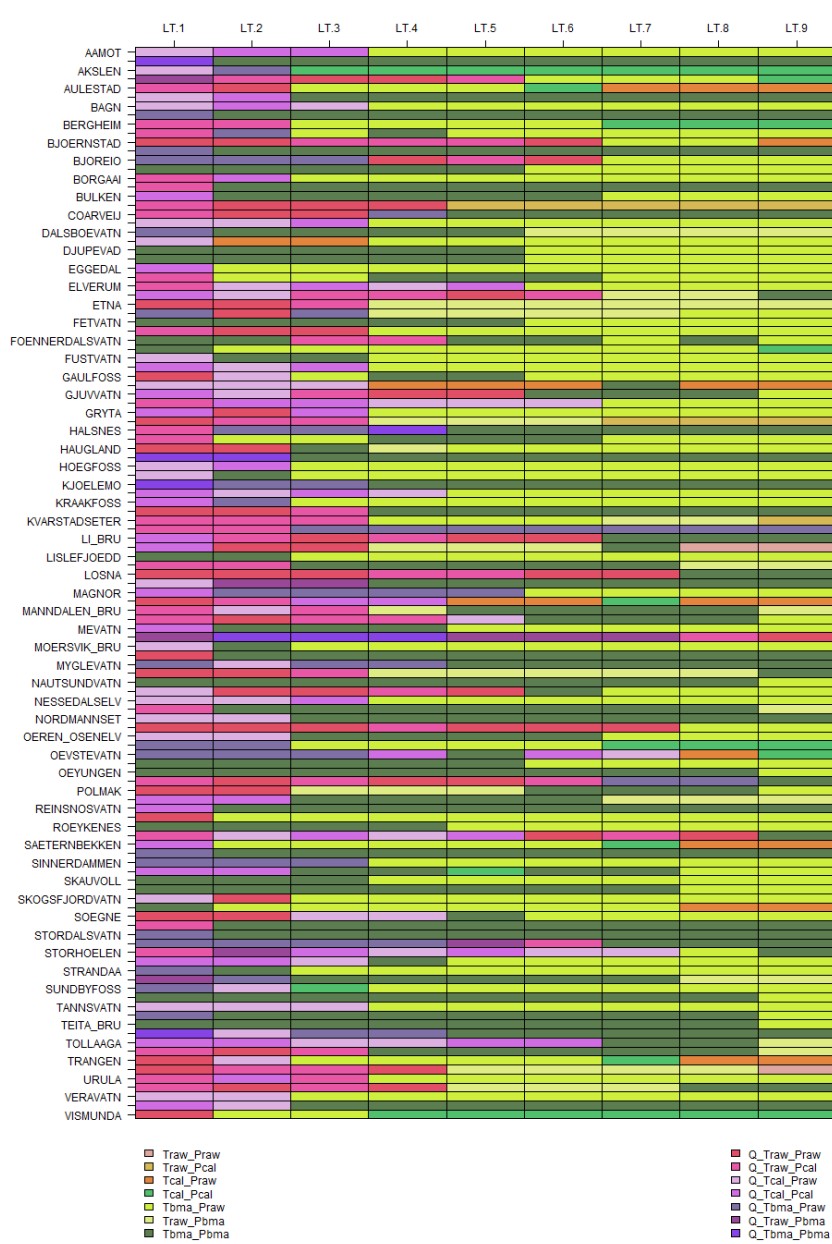

2    A-Figure 3: All data used to evaluate the best CRPS achieved by applied processing schemes, shown for all catchments

3    and lead times. The color in each cell represent the processing scheme with the best CRPS score. Summary of the results

4    shown in A-Figure 4.
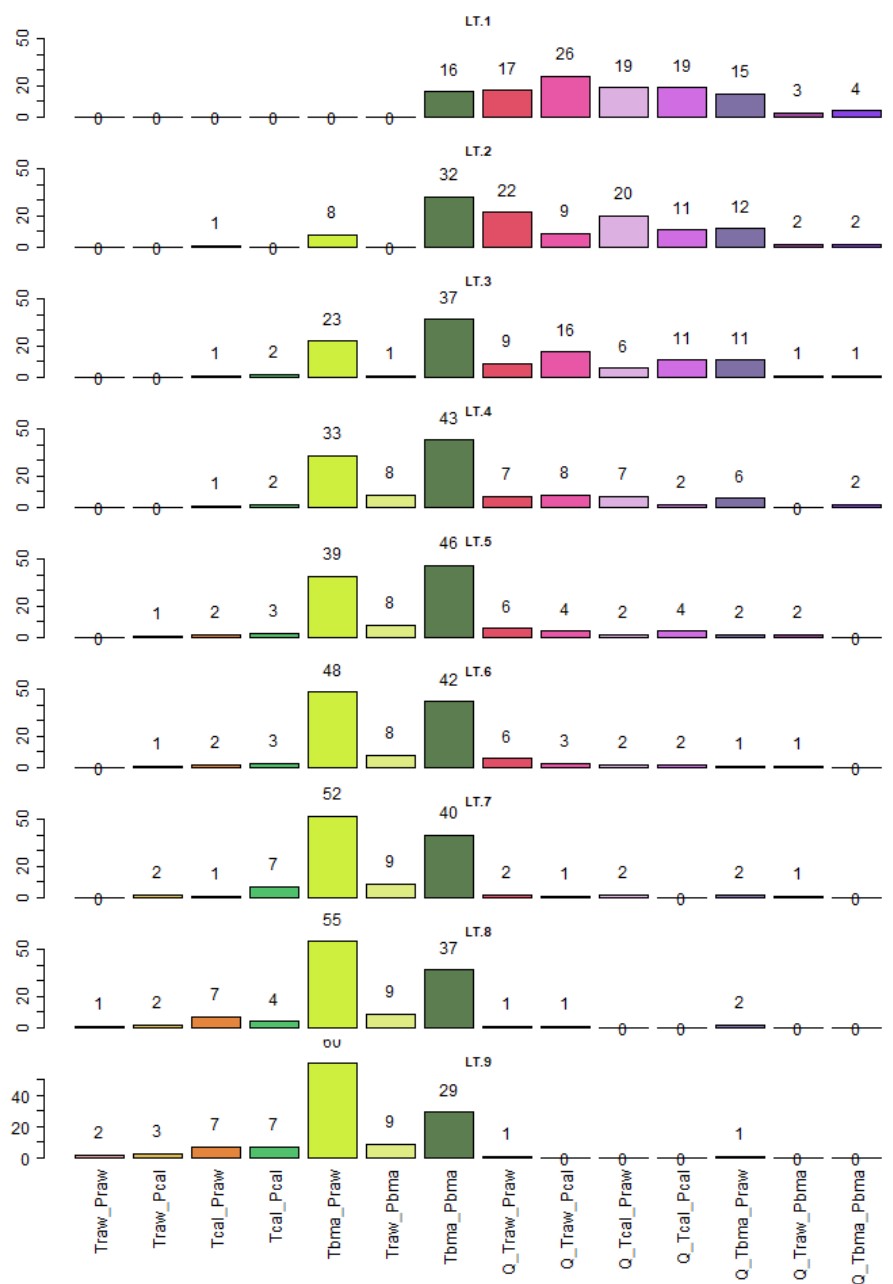

2     **A-Figure 4: Summary of Figure 3. Evaluation applied to all data. Each bar indicates the number of catchments for which**
3     **the specific processing scheme attained the best CRPS score. Lead-times from 1 day (top) to 9 days (bottom).**





**A-Figure 5: Flood dataset used to evaluate the best CRPS achieved by applied processing schemes, shown for all catchments and lead times. The color in each cell represent the processing scheme with the best CRPS score. Summary of the results shown in A-Figure 6.**



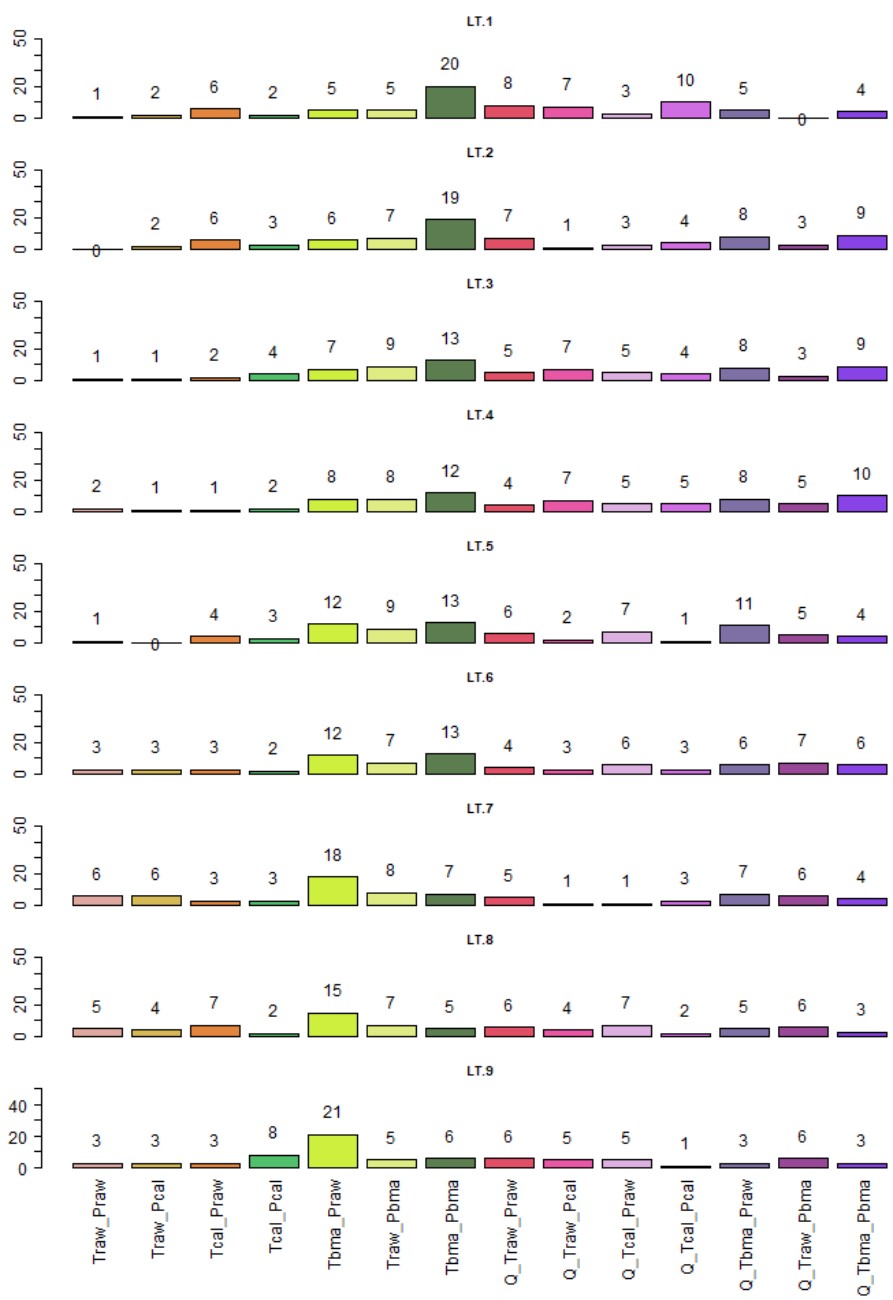

2     **A-Figure 6: Summary of Figure 5. Evaluation applied to the flood dataset. Each bar indicates the number of catchments**

3     **for which the specific processing scheme attained the best CRPS score. Lead-times from 1 day (top) to 9 days (bottom).**



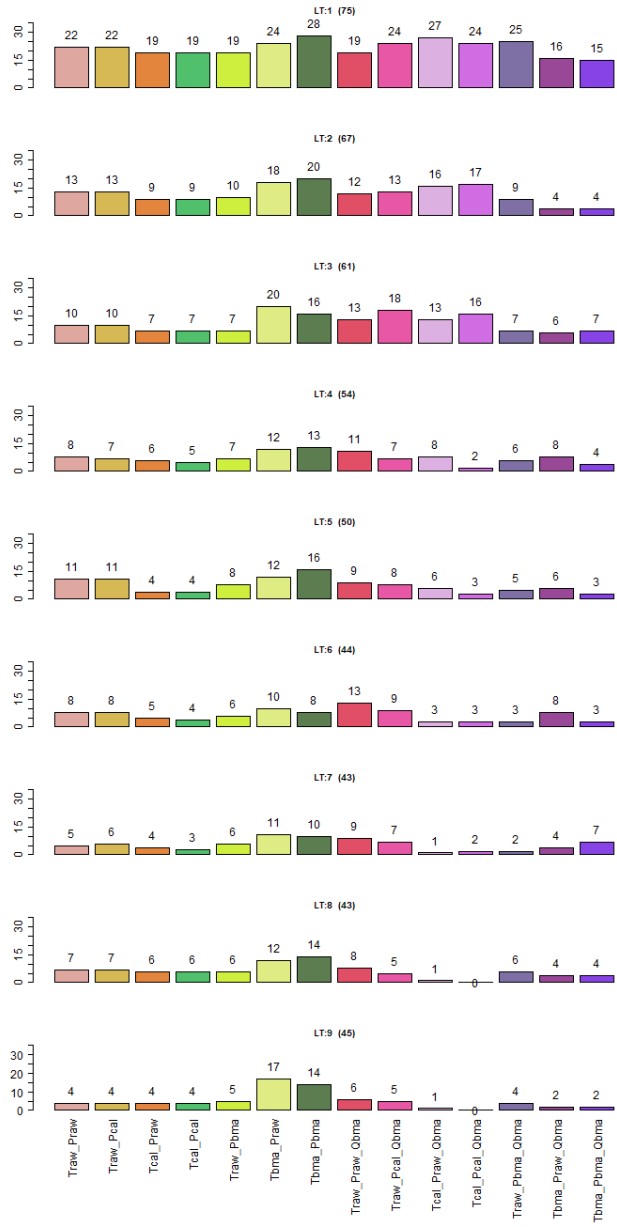

**A-Figure 7: CSI for all catchment (86) where there was either forecasted or observed floods during the 2-year period of**
**the study. In this figure, multiple methods can achieve the criteria for exceeding the flood warning level. The number in**
**parenthesis shows the number of catchments where one or more methods successfully indicates the warning level is**
**exceeded.**