# Peer review of "The benefits of pre- and postprocessing streamflow forecasts for 1 an operational flood-forecasting system of 119 Norwegian 2"

_Hydrology and Earth System Sciences, 2021_

## Author Comment (AC1)

Dear Referee#1

Thank you for the feedback on our article. We appreciate the valuable comments that are helpful in order to improve the manuscript. Replies and corrections are done as follows: the Author response (AR) is marked with red text, while the author's suggestions to corrections (AC) are marked with blue text. All Referee comments are kept in a black. We have split up the text and answered out the referee comments/questions as they occur.

On behalf of the authors
Trine Jahr Hegdahl

AC: We start some general comments to the suggestions by Referee #1. The reviewer suggests us to write a different paper where we focus on evaluating different pre-processing approaches applied to meteorological ensembles for a limited set of catchments.

The reviewer asks why we "needed to look at 119 catchments. Why not pick few catchments and present definite answers to the two questions which Authors have summarized in the conclusion." We think that using a large sample of catchments is the main strength of the paper. In the introduction we have identified this as one of the unique contributions of this paper. One of the main results from our study is that the resulting performance of pre- and processing approaches varies substantially between catchments, and conclusions based on one or a few catchments are therefore not robust.

The reviewer further suggests to "limit the work to only pre-processors, that could be one option". We think that searching for the best combination of pre- and prost processing is an interesting topic. This is also identified as one of the novelties in the introduction and is also the essence of the first research question.

In summary, we want to keep the scope of the paper as it is, i.e., the large sample of 119 catchments and also evaluate the benefit of both pre- and post-processing in different combinations. We will, however, within this scope of the paper, follow up several of the suggestions from the reviewer. We will now continue to address the other issues raised by the reviewer.

Authors reply to Referee #1, 31. March 2021

...

However, from Section 2.3 onwards, I started noticing that there is one line of thinking that Authors are trying to pursue in this manuscript, which may be the serious limitation of the scope of their work.

For instance, why should one use only HBV model, why to use ECMWF ENS data only, how can one interpolates 25 km spatial resolution forecasts to 1 km observation data without any downscaling technique, how Authors estimate the aggregated average values for each catchment, why not one use log-sinh instead of Box-Cox transformation, how justified it is to use Ensemble Coupla-Coupling, why not use Schaake-Shuffle, aren't there other pre- and post- processors than CAL and BMA, etc. etc. There are many such questions which are not addressed here.

AR: There are certainly several research question that could be addressed, but for making a paper with a clear scope, we found it useful to focus on a few research questions and elsewhere make choices based on recommended methods from the literature.

We used the HBV model since this model is part of the operational flood forecasting setup, it is well known and widely used and is hence an interesting model to evaluate We think that adding one more hydrological model would increase the complexity of the study.

ECMWF provides the best medium-range weather forecasts for Norway, and is used operationally, both in weather and flood forecasting. Deterministic forecasts are other options, like ECMWF High resolution forecast, and the short-range forecast by AROME-MetCoOp (not available as an ensemble at the time of the study), or a super ensemble based in the TIGGE archive could have been used. Again, adding more ensemble forecast would increase the complexity of the study.

The 25 km grid was resampled to the 1x1 km grid size using nearest neighbor interpolation combined with elevation adjustment for the temperatures. The BMA and the CAL methods are used to account for the discrepancy between the resolution of the model and the observations and can be regarded as bias-correction or statistical downscaling approaches.

We used the methods described in the text to aggregate the forecasted variables to daily averages.

We have previous studies showing that the Box-Cox transformation performed well for the Norwegian catchments (Engeland et al. 2010) and since this was not the focus of the study, we did not consider a log-sinh transformation.

The Ensemble Copula Coupling and Schaake Shuffle are similar, where the aim is to provide a temporal (and spatial) consistency between the variables that are processed using empirical copulas. The Schaake-Shuffle is often used with a reference to observations/climatology, whereas in the ensemble Coupla Coupling the reference is the ensemble members given by the model and is used to keep the temporal consistency between temperature and precipitation for each of the ensemble members.

There are plenty of pre- and post-processing methods (also mentioned in the manuscript), we did however choose to compare two approaches. One approach is to make use of data provided by a weather service center, where the focus is on improved precipitation and temperature forecasts for larger areas, this is referred to as the gridded approach (CAL). The alternative approach (BMA) is adaptive, and the calibration is catchment specific. We did not include more techniques in this study.

There are several other interesting approaches, methods, and questions to be asked. We think, however, that including several models, forecasts, and methods will increase the complexity of the study and not be beneficial to improve the clarity of this study.

AC: We will address each of these issues raised by the reviewer in a revised manuscript, by further explain the choices we made with respect to datasets, models, transformations etc. and refer to recommendations from literature when suitable.

In other words, I couldn't find what is novel here, knowing very well that there are several papers on this topic already published. Practically, every month we find new publication on pre- or post-processing in different journals.

AR: We argue that the number of catchments and the inclusion of pre- and postprocessing for both daily streamflow, and floods are a novelty. However, if we were to reduce the study to a few catchments, and only focus on preprocessing we find that the novelty will vanish.

Why not pick few catchments and present definite answers to the two questions which Authors have summarized in the conclusion. Since the study tried to capture so many different aspects, physiography, seasonal, snow-melt vs. rainfall based flood, etc. etc. that lead to Authors having fairly standard conclusions.

Instead, I would look into few aspects but with rigorous analysis and try to derive some conclusions which can actually benefit the hydro-meteorological forecast community, not only in Norway but other places also.

AR: One of the main results from this study is that it is very difficult to be conclusive about transferring information from a single study. We do however provide some insights that is important to consider for the usefulness of processing.

It is very important to have plots which can be interpreted easily.
In this manuscript, almost all the results are shown through box-plots. A set of time series plots showing how good the pre- and post-processors are improving the forecast would be highly beneficial.

AC: We can provide time series plots for selected catchments, showing the median forecast for each method at e.g., lead time 1, 5, and 9 days for the study period.

Similarly, to show the improved flood forecasting, a time series plot would make things very clear.
But I can only imagine the difficulty one would face in summarizing the results of 119 catchments, 51 ensembles, 9 lead time etc.

AR: We will do an effort to find alternative ways of presenting the results from this study.

Therefore, a small number of catchments from different parts of the country may be the way forward. In one sense, Authors are already doing this by summarizing results of only 6 catchments.

Then please limit the work to only pre-processors, that could be one option. In summary, Authors have to find a way to focus on novelty of this study rather than trying to cover all possible aspects on this topic.

AR: Only focus on preprocessors for the flooding events.

**Citation**: https://doi.org/10.5194/hess-2021-13-RC1

Reference:
Engeland, K., Renard, B., Steinsland, I., and Kolberg, S.: Evaluation of statistical models for forecast errors from the HBV model. Journal of Hydrology, 384(1), 142-155, 2010.

---

## Author Comment (AC2)

Authors reply to Referee#2, 20 April 2021

Dear Referee#2

Thank you for the feedback on our article. We appreciate the valuable comments that are helpful in order to improve the manuscript. Replies and corrections are done as follows: the Author response (AR) is marked with red text, while the author's suggestions to corrections (AC) are marked with blue text. All Referee comments are kept in a black.

On behalf of the authors
Trine Jahr Hegdahl

This review is for Manuscript No.: hess-2021-13, entitled "The benefits of pre- and postprocessing streamflow forecasts for an operational flood-forecasting system of 119 Norwegian catchments", authored by Trine J. Hegdahl, Kolbjørn Engeland, Ingelin Steinsland, and Andrew Singleton. With this manuscript, the authors examine the benefits of preprocessing metrological forcing and/or postprocessing streamflow forecast in improving the quality of ensemble streamflow forecasts. I believe the topic is of interest to the hydrometeorological community. However, I have major comments that needs to be addressed before this manuscript is ready for publication.

1. Abstract is quite long. For one, third statement (First Paragraph) convey the same message as the First and Second statements. Results in the Abstract Section need to be well summarized (focus on important findings, rather than mentioning all of your results).

   AC: We will avoid repetitions and reduce the length of the abstract. Further, we will focus on the main findings of the study.

2. The authors are using different training period (2014, 2006-2011, 45 previous days) for different preprocessing schemes. To ensure a fair comparison of methods, they need to each be assessed using an equivalent systematic method to determine the optimal training set for each method.

   AR: In the study we used the processing method used by the Norwegian meteorological institute for temperature and precipitation, including the set of tuned parameters in the pre-processing model (i.e. grid calibration). In addition, we performed our own processing using BMA. The grid-calibration and the BMA calibration differs in several ways, and reflects the difference in processing apporaches:

   - BMA is tuned to each catchment individually whereas the grid calibration is tuned globally to all Norway
   - BMA uses a sliding window of 45 days for training the model, whereas the grid-calibration used previous years to train the model. The parameters of the BMA model change for each issue date, whereas the grid calibration has parameters that depend on season.

   The idea of using an already exciting pre-processing method, was two-fold. Firstly, we wanted to assess if the existing grid-calibration improved the forecast compared to using the raw ensembles. Secondly, we wanted to assess the potential improvement by catchment specific calibration of the post-processing model. The results from this study show that the grid calibration, even though not optimized for the catchments, improved the skill of the operational hydrological forecasts for most cases. However, using BMA where the forecasts were tuned to each individual catchment gave the best performance.

To define the optimal training length for BMA, we chose to use the same training length for P, T and Q. This was done to ensure a fair comparison of the methods. We did however see that optimal training length depend on catchment, lead time and variable. The dataset is as pointed out limited, and we assume that optimal training length to a large degree will vary depending on the availability and representativeness of events in the training period.

AC: We think that since the two processing approaches differs in many aspects, it is challenging to discuss and explain why the outcome is different. To simplify both figures presenting the results and the discussion, we will exclude the results from the grid-calibration and only use BMA.

3. The authors are preprocessing and/or postprocessing the flood events. How realistic is it to preprocess (postprocess) large and rare precipitation events (flood events) by using just 45 previous days of training period? For flood events, it seems like the longer training period (multiple years, if available) is generally advantageous.

AR: We agree that the choice of training data for the pre- and processing algorithm is a critical issue. In this study we use a sliding window, as suggested originally by Rafetry et al. (2005) and later on used in several papers where the BMA approach is used for post-processing (e.g. Sloughter et al., 2007, Li et al., 2020). Typical sizes for the training window are 30 to 60 days. One argument for using a sliding window for training, is that the calibration adjusts to seasonal variations in model biases and easily adjusts to new model versions. In Raftery et al (2005) a window size of 60 days was used. Sloughter et al. (2007) used a window size of 30 days and found that increasing the training period beyond that did not further improve the sill of the forecasts. The choice of window size is a trade of between having enough data for estimation and obtaining a flexible post-processing approach that adapts to the most recent data. We tested several training lengths (A-Figure 1 and A-Figure 2 in supplement), and we needed at least 45 days to have enough days with no-zero precipitation, and we used the same training length for all forecasted variables. We acknowledge that pre- and post-processing extreme precipitation and floods using BMA is difficult. The reason is that the forecast performance of extreme event might be very different from the forecast performance for the training window. We think that an alternative approach to select training data might be to look for similar events.

AC: We will evaluate how sensitive our results are to the size of the training window and summarize the outcome.  We will add a paragraph in the discussion about the choice of training data for the BMA method, the size of the sliding window and alternative ways to select training data.

[Figure]

*Figure 1Example of CRPS (vertical axis, small is better) for different BMA training lengths (30, 45 and 60 days) for precipitation. Lead times on the horizontal axis. The upper thinner lines are evaluated on floods only. Catchments from left: Bulken, Røykenes and Møska.*

4. The authors use cubed root transformation in their BMA model for precipitation. How were this transformation chosen? Did the authors choose cubed root without testing alternative transformations?

   AR: Different power law transformations of precipitation within a BMA framework has been investigated in several papers, and the cube root transformation has been shown to give the best result  (see  e.g. Sloughter et al, 2007, and Li et al., 2020) and is often used as a standard transformation when processing precipitation. This is also the transformation tested and used by The Norwegian meteorological institute Norway (personal communication). In this study we chose to use this standard transformation.

   AC: We will add a sentence that inform about alterative power law transformations and use the papers by Sloughter et al (2007) and Li et al., (2020) to explain why the cubic root transformation was used.

5. Most of the Figures (Figure 5 - Figure 14; and Appendix-Figures) are difficult to follow. This really creates difficulty in reading the Result Section. Please make all the Figures simple and easy to follow.

   AR: It is challenging to find a good way to summarize results from a large dataset, and we will look for alternative approaches to present the results.

   AC: We will simplify the figures by reducing number of lead times, (present only every second lead time) and methods (present 7 instead of 14) presented, by

excluding the grid-calibration. We will keep the box-plots in Figures 6,7,10 and 11, but make the interpretation of the results simpler, and look at how we can e.g., change the colors to improve the figures.

**Citation**: https://doi.org/10.5194/hess-2021-13-RC2

Li, X, Chen, J, Xu, C-Y, Chen, H, Guo, S. Intercomparison of multiple statistical methods in post-processing ensemble precipitation and temperature forecasts. Meteorol Appl. 2020; 27:e1935. https://doi-org.ezproxy.uio.no/10.1002/met.1935

Raftery, A. E., Gneiting, T., Balabdaoui, F. and Polakowski, M.: Using Bayesian model averaging to calibrate 6 forecast ensembles. Monthly Weather Review, 133(5), 1155-1174, 2005.